# Components Beat Patches: Eigenvector Masking for Visual Representation Learning

## Abstract

Masked Image Modeling has gained prominence as a powerful self-supervised learning approach for visual representation learning by reconstructing masked-out patches of pixels. However, the use of random spatial masking can lead to failure cases in which the learned features are not predictive of downstream labels. In this work, we introduce a novel masking strategy that targets principal components instead of image patches. The learning task then amounts to reconstructing the information of masked-out principal components. The principal components of a dataset contain more global information than patches, such that the information shared between the masked input and the reconstruction target should involve more high-level variables of interest. This property allows principal components to offer a more meaningful masking space, which manifests in improved quality of the learned representations. We provide empirical evidence across natural and medical datasets and demonstrate substantial improvements in image classification tasks. Our method thus offers a simple and robust data-driven alternative to traditional Masked Image Modelling approaches.

## 1 Introduction

Masked Image Modeling (MIM; Pathak et al., 2016; He et al., 2021; Bao et al., 2022; Xie et al., 2022) draws inspiration from masked language modeling (e.g., BERT; Devlin, 2018), where parts of a sentence are masked, and a model has to learn to predict the missing words. Similarly, in MIM, portions of an image are masked out, and a model has to reconstruct the missing parts from the visible ones. To do well at this task, it is thought that the model is forced to learn a meaningful representation of the visual content in the process (Kong et al., 2023). Empirically, this approach indeed tends to produce representations that perform particularly well when fine-tuned on various downstream tasks, such as image classification and semantic segmentation (He et al., 2021).

The MIM paradigm has led to significant advances in the field of self-supervised learning (SSL) of visual representations (Pathak et al., 2016; Zhou et al., 2021; He et al., 2021; Bao et al., 2022; Xie et al., 2022; Baevski et al., 2022; Dong et al., 2023) and has been particularly effective when combined with Vision Transformers (ViT; Dosovitskiy et al., 2021). A prominent example of this is the Masked Autoencoder (MAE; He et al., 2021), which consists of two core components: a ViT encoder-decoder architecture and a masking strategy that randomly selects a fixed ratio of square image patches. The encoder processes the visible patches (along with their positional embeddings) into a representation that the decoder can use to accurately reconstruct the masked-out content.

While the inner workings of MIM in general, and MAEs in particular, remain under-explored and poorly understood (Zhang et al., 2022; Yue et al., 2023), Kong et al. (2023) recently suggested a potential explanation from a latent variable model perspective: by splitting an image into two parts and asking the model to predict one from the other, MAEs are compelled to pick up on any information shared between the two that is helpful to solve the image modeling task. If the partition into parts (i.e., the masking strategy) is chosen *carefully*, this shared information will include high-level latent variables, such as object class. Since solving common downstream tasks with a simple (e.g., linear) predictor precisely requires identifying such high-level information, this offers a possible explanation for the observed effectiveness of MIM/MAE representations.

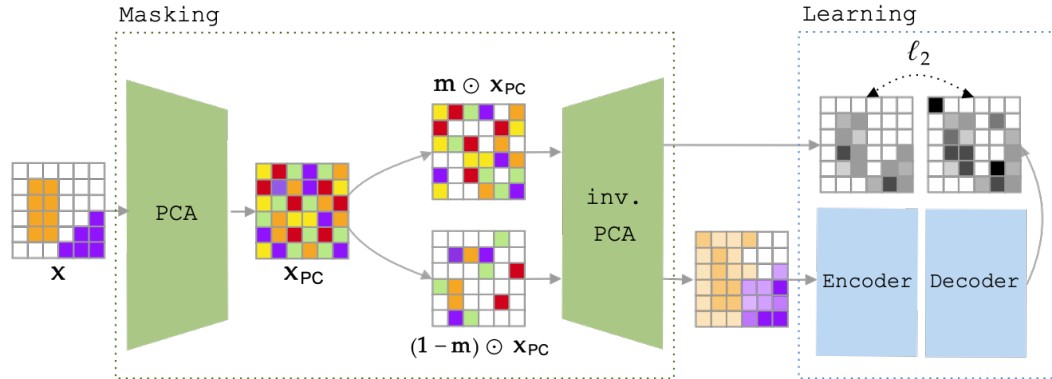

Figure 1: **Overview of the Principled Masked Autoencoder (PMAE).** A Principled Masked Autoencoder (PMAE) differs from a vanilla MAE by performing the masking in the space of principal components $\mathbf{x}_{\text{PC}} = \text{PCA}(\mathbf{x})$ rather than in the observation space. The masked principal component representations $\mathbf{m} \odot \mathbf{x}_{\text{PC}}$ and $(\mathbf{1} - \mathbf{m}) \odot \mathbf{x}_{\text{PC}}$ are then projected back into the observation space and serve as the reconstruction target and input for an encoder-decoder architecture, respectively.

With this in mind, it seems reasonable to ask: *Is patch-wise masking in pixel space really the best strategy for MIM?* In natural language, each word in a sentence tends to carry semantic information, and the information shared between sets of words often conveys the general message of the sentence. However, this does not necessarily apply to the visual domain. In images, individual pixels (and sometimes even entire patches) may contain information redundant that of other pixels. For example, many background pixels will often be identical. Moreover, objects can be masked out completely, such that any information about them is lost and reconstruction becomes impossible (see, e.g., Fig. 2, left). The widely adopted strategy of masking image patches (i.e., *spatial masking*) may thus be sub-optimal and lead to representations that capture information that is irrelevant for downstream tasks of interest.

Some works have thus sought to devise better masking strategies by relying on auxiliary information such as (learned or inferred) image segmentations (Li et al., 2021; Kakogeorgiou et al., 2022; Shi et al., 2022). Without prior knowledge or more complex training pipelines to identify the structure of an image, randomly masking a fixed proportion of patches remains the default practice. However, relying on this strategy assumes—rather unrealistically—that the information shared between any random partition of patches naturally aligns with high-level variables of interest (Kong et al., 2023).

In this work, we introduce *a new data-driven masking strategy for MIM*. Rather than working directly in pixel space, we propose to first project images into a latent space and then perform the masking on the transformed data. Specifically, we opt for off-the-shelf data projections using principal component analysis (PCA) and mask a random subset of the principal components. We refer to the resulting method as Principled Masked Autoencoder (PMAE), see Fig. 1 for an overview.

We argue that the space of principal components constitutes a more meaningful domain for masking, since it allows for partitioning the information in an image based on global features rather than local patches of pixels. This helps overcome some of the aforementioned failure modes of spatial masking and results in learning more useful high-level representations. Indeed, by masking globally rather than locally, we avoid scenarios where the masked out and visible information are either too strongly correlated (where visible information is redundant with what is masked out) or too weakly correlated (where visible information fails to predict what is masked out). Recent work has also highlighted the beneficial partitioning of image information by PCA: Balestriero & LeCun (2024) demonstrate that low-eigenvalue components capture features crucial for common downstream tasks (see also Fig. 7); and Chen et al. (2024b) highlight the importance of the space in which image distortions are applied, referring to PCA as a valuable transformation to consider. To the best of our knowledge, our work is the first to leverage such insights to devise a simple, robust, and effective data-driven alternative to pixel-space-masking in MIM.

We evaluate PMAE in experiments on natural and medical image datasets where it consistently yields substantial performance gains over spatial masking. For linear probing experiments, we report an average performance gain of $26\%$ over the widely adopted strategy of masking out $75\%$ of images

Figure 2: **(Left) Masking in pixel space.** TinyImageNet sample (left) with a random spatial mask *partially* removing relevant information (middle) and a random spatial mask removing *all* semantic information (right). The latter constitutes an example in which MIM would fail to learn useful representations. **(Right) MedMNIST datasets .** Example images from the (from left to right) DermaMNIST, PathMNIST, and BloodMNIST datasets used for image classification (Yang et al., 2023).

patches. Interestingly, we find that without any hyperparameter tuning, PMAE outperforms spatial masking with optimal hyperparameters in all but one dataset. These results support the belief that modeling masked-out principal components facilitates the learning of meaningful representations.

## 2 BACKGROUND

**Principal Component Analysis.** Principal Component Analysis (PCA; Pearson, 1901; Hotelling, 1933) aims to expose data components that exhibit high variation. Given a centered data matrix $\mathbf{X} \in \mathbb{R}^{N \times D}$ consisting of $N$ observations of dimension $D$, PCA iteratively seeks weight vectors $\mathbf{v}_l \in \mathbb{R}^D$ for $l = 1, ..., L \leq D$, called principal components (PC), which maximize the variance of the linear projection $\mathbf{X}\mathbf{v}_l$ of the columns of $\mathbf{X}$, subject to being orthogonal to the previously found $\mathbf{v}_1, ..., \mathbf{v}_{l-1}$ and of unit-length. The solution to this problem is given by the eigenvalue decomposition of the empirical covariance matrix $\mathbf{\Sigma} := \mathbf{X}^\top \mathbf{X}$, i.e., $\mathbf{\Sigma} = \mathbf{V}\mathbf{\Lambda}\mathbf{V}^\top$, where $\mathbf{\Lambda} = \mathrm{diag}(\lambda_1, ..., \lambda_D)$ contains the ordered eigenvalues $\lambda_1 > ... > \lambda_D$, and the corresponding eigenvectors are the columns of $\mathbf{V} \in \mathbb{R}^{D \times D}$. In other words, the first $L$ principal components are given by the eigenvectors of $\mathbf{X}^\top \mathbf{X}$ corresponding to the $L$ largest eigenvalues. Eigenvectors with higher eigenvalues can thus be seen as capturing the dominant modes of variation in the data. Further, the variance explained by each principal component can be shown to be proportional to its corresponding eigenvalue $\lambda_l$. Whereas PCA is often used with $L < M$ for dimensionality reduction, we will focus on the lossless case with $L = M$ throughout. The projection $\mathbf{X}_{\mathrm{PC}}$ of $\mathbf{X}$ onto its principal components ("*into PC space*") is then given by $\mathbf{X}_{\mathrm{PC}} = \mathbf{X}\mathbf{V}$, and the inverse transformation back into observation space by $\mathbf{X} = \mathbf{X}_{\mathrm{PC}}\mathbf{V}^\top$. As $\mathbf{V}$ is an orthonormal basis, the columns of $\mathbf{X}_{\mathrm{PC}}$ *might* be statistically independent if and only if the columns of $\mathbf{X}$ are themselves *linear* combinations of independent factors. In this work, we focused our exploration on the natural image domain, where the assumption that each pixel is a *non-linear* combination of a set of independent factors is widely considered realistic. Please refer to Appx. A.1 for further details regarding PCA.

**Representation Learning.** Representation learning (Bengio et al., 2013) aims at learning an embedding function $f : \mathbf{x} \mapsto \mathbf{z}$, which maps data observation $\mathbf{x} \in \mathbb{R}^D$ to latent representation $\mathbf{z} \in \mathbb{R}^K$. These latent representations are meant to capture some of the explanatory factors underlying the data, thus making $\mathbf{z}$ well-suited for use in downstream tasks such as predicting $y$ (e.g., the class or location of objects), often thought of as a being a function of the data's explanatory variables.

**Masked Image Modelling.** Prominent approaches to representation learning in the image domain rely on the masked image modeling paradigm (Pathak et al., 2016; Zhou et al., 2021; He et al., 2021; Bao et al., 2022). We choose the widely adopted Masked Autoencoder (He et al., 2021) as a representative of MIM. The encoder-decoder architecture in MAE allows the model to reconstruct masked portions of data observations by leveraging the learned latent representations:

$$\mathcal{L}_{\mathrm{MAE}}(\mathbf{x}, \mathbf{m}; \boldsymbol{\theta}, \boldsymbol{\phi}) = \|\mathbf{m}^{\mathrm{c}} \odot g_{\boldsymbol{\theta}}\left(f_{\boldsymbol{\phi}}\left(\mathbf{m} \odot \mathbf{x}\right)\right) - \mathbf{m}^{\mathrm{c}} \odot \mathbf{x}\|_2^2, \tag{2.1}$$

where $\mathbf{x}$ is a data observation and $\mathbf{m}$ and $\mathbf{m}^{\mathrm{c}} = (\mathbf{1} - \mathbf{m})$ are complementary binary masks used to extract the visible and masked-out parts of the data, respectively. The embedding function $f$, parametrized by $\boldsymbol{\phi}$, encodes the visible portions of the input together with their positional embed-

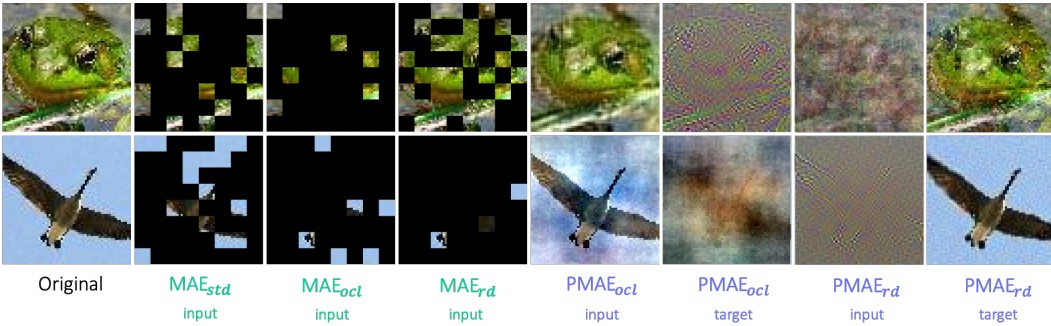

Figure 3: **Mask Design Strategies.** An overview of the different mask design strategies used in our experimental setup: spatial masking (*green*) and principal component masking (*blue*). *std* refers to the standard approach of masking out $75\%$ of image patches, *ocl* denotes masking with the optimal masking ratio, *rd* represents a randomized strategy where the masking ratio is randomly sampled for each batch, and *target* refers to the reconstruction target;

dings, while the decoder function $g$, parametrized by $\boldsymbol{\theta}$, reconstructs the missing parts from their positional embeddings and the latent representation $\boldsymbol{z} = f_{\boldsymbol{\phi}}\left(\mathbf{m} \odot \mathbf{x}\right)$.

The binary mask $\mathbf{m} = \{m^i\}_{i=1}^{D}$ partitions the $D$ pixels into two disjoint sets of $(1-r)D$ visible and $rD$ masked out pixels, where $r$ is referred to as the *masking ratio*. Patch-wise masking, which masks patches of pixels instead of individual pixels, introduces the patch size as an additional hyperparameter. $r$ then defines the amount of patches to be masked out. Prior work has relied on hyperparameter sweeps to identify the masking ratio and patch size that optimize downstream performance (He et al., 2021; Zhang et al., 2022). These efforts have led to the widely adopted approach of masking out $75\%(r=0.75)$ of image patches.

**Intuition behind MIM.** Despite the MIM paradigm with random spatial masking producing strong results on representation learning benchmarks (Dong et al., 2023), it is based on the rather unrealistic assumption that for any partition of an image's patches into two disjoint sets, the information shared by these sets contains $y$ (Kong et al., 2023). In Fig. 2, we observe that while some masks (middle image) may allow shared information to include the object type and corresponding class label, there are many partitions where predicting the label (e.g., the class label *"goose"* in Fig. 2) from the visible patches is almost as uncertain as a random guess (left image). Moreover, even for well-designed masks, a substantial proportion of masked-out patches contain information redundant with visible pixels. We conjecture that, as a result, Masked Image Modeling with spatial masking leads to a suboptimal learning approach that is misaligned with common downstream tasks, is characterized by slow convergence and suffers from high sensitivity to hyperparameters, as suggested by prior work (He et al., 2021; Balestriero & LeCun, 2024) and confirmed in Section 5.

## 3   ROBUST MASKED IMAGE MODELLING

To address the challenges presented in Section 2, we propose *Principled Masked Autoencoders* (PMAE). PMAE builds on the MIM learning paradigm, but differs from prior approaches by performing the masking operation in a learned latent space, resulting in the following objective:

$$\mathcal{L}_{\mathrm{PMAE}}(\mathbf{x}, \mathbf{m}; \boldsymbol{\theta}, \boldsymbol{\phi}) = \left\| g_{\boldsymbol{\theta}}\left(f_{\boldsymbol{\phi}}\left(h\left(\mathbf{m}, \mathbf{x}\right)\right)\right) - h(\mathbf{m}^{\mathrm{c}}, \mathbf{x})\right\|_2^2, \tag{3.1}$$

where $h(\mathbf{m}, \mathbf{x}) = t^{-1}(\mathbf{m} \odot t(\mathbf{x})) = t^{-1}(\mathbf{m} \odot \mathbf{x}_{\mathrm{PC}})$, and $t$ is an invertible function, $t : \mathbb{R}^D \to \mathbb{R}^D$ mapping the input $\mathbf{x}$ to a representation space $\mathbf{x}_{\mathrm{PC}} = t(\mathbf{x})$. Eq. (3.1) and Eq. (2.1) differ in that the masking operates within the latent space. Similar to Eq. (2.1), the embedding function $f$, parametrized by $\boldsymbol{\phi}$, encodes the visible portions of the input together, while the decoder function $g$, parametrized by $\boldsymbol{\theta}$, reconstructs the missing parts.

Note that while the masking is performed in latent space, Vision Transformers (Dosovitskiy et al., 2021) generally perform remarkably well in pixel space, and we thus keep the reconstruction task in

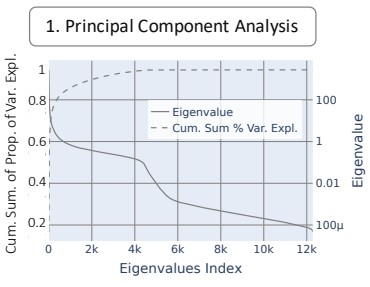 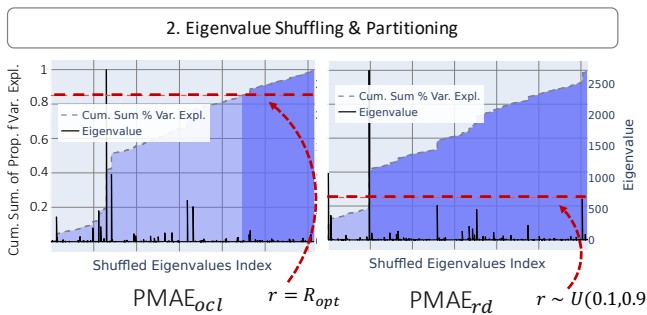

Figure 4: **Mask Design in PMAE.** 1. Principal Component Analysis is performed 2. For each batch, principal components are randomly shuffled and a subset is selected to construct the input (light blue), while the remaining components serve to construct the reconstruction target (dark blue). In $\text{PMAE}_{ocl}$, the input components are chosen to explain $((1 - r) \times 100)\%$ of the data's variance. $r$ is optimized for downstream performance, here $R_{\text{opt}}$ is set to $0.15$. In $\text{PMAE}_{rd}$, the input explain between 10% and 90% of the variance. $r$ is sampled from $U(0.1, 0.9)$ for each batch independently.

the observation space. Consequently, after the masking in latent space, $t^{-1}$ projects $\mathbf{x}_{\text{PC}}$ back to the observation space. Fig. 1 provides a visual overview of our approach.

**Intuition behind PMAE.** In contrast to the traditional spatial masking presented in Section 2, an appropriate function $t$ can encourage information shared between visible and masked-out information to contain $y$. More specifically, if the latent space captures unique global information in each dimension, masking any of these dimensions retains information about all parts of the image. Hence, Eq. (3.1) allows us to learn more meaningful representations for a suitable choice of $t$.

The appeal of the proposed approach then boils down to finding $t$. While there may be many appropriate choices for $t$, we found that applying PCA and projecting samples using the resulting principal components is a suitable choice for the latent space. In particular, each dimension captures specific factors of variation observed within the dataset and is typically tied to global features as shown in Fig. 7. Masking one factor of variation thus prevents us from completely removing all information about variables of interest within a sample, as most principal components will retain some information about them. We will present the positive impact of this reasoning in Section 5, where we compare principal component and observation space masking empirically. In Section 6, we will provide additional intuition as to why PCA leads to a suitable masking strategy.

## 4 EXPERIMENTAL SETUP

We now outline the setup used to validate PMAE. Our experiments follow the evaluation proposed by He et al. (2021), ensuring the comparability of results between our PMAE and baselines. Details regarding this experimental setup and computational training costs can be found in Appx. A.2.

**Mask Design.** State-of-the-art MIM models, like MAE (He et al., 2021), typically employ random image patch masking, which serves as our baseline. Based on ablation studies from He et al. (2021), the standard practice involves masking out 75% of image patches (denoted as $\text{MAE}_{\text{std}}$). We also examine an oracle-based masking strategy (denoted as $\text{MAE}_{\text{ocl}}$), where the masking ratio is fine-tuned to optimize linear probing downstream performance. This setting serves as an upper-bound to the downstream performance. Additionally, we introduce a randomized masking approach, $\text{MAE}_{\text{rd}}$, in which the masking ratio is independently sampled for each batch, within a range of 10% to 90% of image patches being masked out. This strategy is exempt from any hyperparameter tuning and offers insights into the downstream performance when using suboptimal masking hyperparameters.

A similar approach is applied to PMAE, where we consider both oracle ($\text{PMAE}_{\text{ocl}}$) and randomized ($\text{PMAE}_{\text{rd}}$) masking strategies. In PMAE, we define the masking ratio as the proportion of data variance to be masked out. Indeed, while for spatial masking, the masking ratio represents the proportion of patches to be masked out, with PCA, each principal component accounts for a percentage

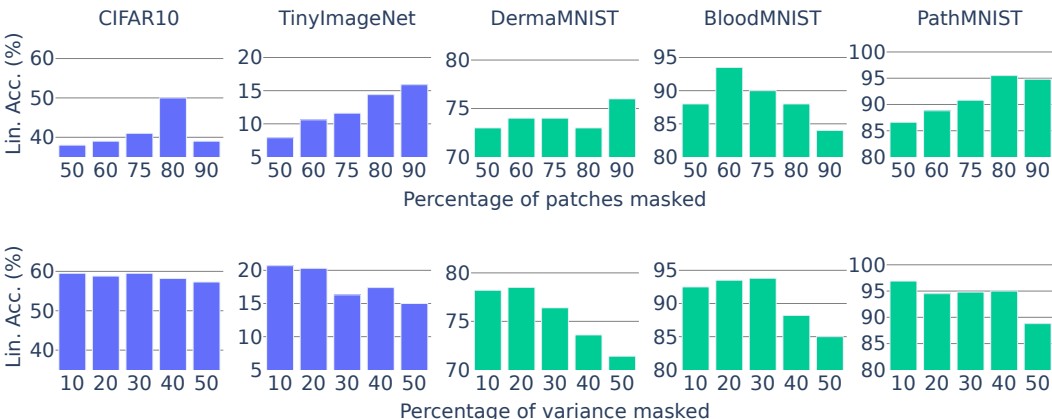

Figure 5: **Impact of the Masking Ratio.** MAE (top) and PMAE (bottom) linear probing accuracy for varying masking ratios. The masking ratio is a sensitive and data-dependent hyper-parameter. While for MAE a clear masking guideline is hard to extract, for PMAE we observe a close to optimal performance across datasets for 10 to 20% of the data variance masked.

of the data variance. In the oracle approach, we define the optimal percentage of *variance* to be masked based on downstream performance on a held-out dataset. In the randomized strategy, we simply ensure at least 10% and at most 90% of the data variance is masked out. This percentage is independently sampled for each batch.

Fig. 3 provides examples of the images obtained from these masking strategies. Note that Fig. 3 helps visualise how MAE masks out patches of local information while PMAE operates globally over the image. Fig. 4 provides further practical insights into how the masking of principal components is performed for the oracle and randomized settings. After PCA, the order of principal components is randomly shuffled for each batch. The components are then partitioned into two disjoint sets, explaining $100\times(1-r)\%$ and $100\times r\%$ of the variance for masking ratio, $r$. The masking ratio is fixed for $\mathrm{PMAE_{ocl}}$ and independently sampled for each batch from the $[0.1, 0.9]$ range in $\mathrm{PMAE_{rd}}$. Both partitions are then projected back to the observation space to serve as input and reconstruction target.

**Training & Evaluation.** We train ViT-T/8 encoder and decoder backbones (Dosovitskiy et al., 2021). We fix the patch size to $8\times8$ pixels. Following common practice, we use image flipping random image cropping as data augmentation (He et al., 2021). We train representations for 800 epochs and provide an overview of the evolution of performance across training in Appx. A.7. We then evaluate learned representations on image classification using a linear probe and multi-layer perceptron (MLP) classifier on top of the encoder's output [CLS] token which is frozen. Following He et al. (2021), we fix the training duration of the linear and MLP probes to 100 epochs. Appx. A.7 also reports downstream performance obtained with a $k$-NN classifier.

## 5 RESULTS

In this section, we will outline and analyze the empirical advantages of PMAE compared to standard MAEs in image classification tasks. Specifically, we provide evidence that masking within the space of principal components facilitates the learning of discriminative features, resulting in improved performance on downstream tasks. Our findings are supported by empirical evidence across diverse datasets, including two natural image datasets of $32\times32$ and $64\times64$ image resolutions, and three medical datasets taken from MedMNIST (Yang et al., 2023) of $64\times64$ image resolution (see Fig. 2 for example MedMNIST images).

Tab. 1 presents the classification accuracy using both a linear probe and a MLP classifier. Across datasets, we observe substantial improvements with $\mathrm{PMAE_{ocl}}$ in linear probing compared to the standard $\mathrm{MAE_{std}}$, with an average increase of 26% (+10 percentage points). Additionally, $\mathrm{PMAE_{ocl}}$ outperforms $\mathrm{MAE_{ocl}}$ by 10.9% (+6.6 percentage points). We see similar gains with the randomized

Table 1: Linear and MLP probe top-1% accuracy for CIFAR10, TinyImageNet and MedMNIST datasets for random masking in pixel (MAE) and principal component (PMAE) space with the standard 75% masking ratio (std), oracles (ocl) and randomized masking ratios (rd). * refers to ours.

|  |  | CIFAR10 | TinyImageNet | DermaMNIST | BloodMNIST | PathMNIST |
|---|---|---|---|---|---|---|
| **Linear** | $MAE_{std}$ | 41.7 | 11.5 | 72.4 | 73.4 | 83.4 |
|  | $MAE_{ocl}$ | 50.7 | 15.5 | 73.7 | 78.6 | 86.4 |
|  | $PMAE^*_{ocl}$ | **55.1** | **17.4** | **77.4** | **91.0** | **97.0** |
|  | $MAE_{rd}$ | 41.9 | 7.5 | 72.4 | 83.2 | 85.6 |
|  | $PMAE^*_{rd}$ | **56.0** | **15.1** | **74.5** | **85.9** | **87.5** |
| **MLP** | $MAE_{std}$ | 34.0 | 15.5 | 72.2 | 68.6 | 92.6 |
|  | $MAE_{ocl}$ | 55.2 | **22.2** | 74.4 | 75.8 | 95.1 |
|  | $PMAE^*_{ocl}$ | **61.5** | 22.1 | **79.6** | **91.0** | **98.8** |
|  | $MAE_{rd}$ | 38.5 | 11.6 | 66.9 | 70.6 | 95.7 |
|  | $PMAE^*_{rd}$ | **62.2** | **19.5** | **75.3** | **84.4** | **97.0** |

hyperparameter strategy. PMAE consistently outperforms MAE across all datasets, yielding an average performance increase of 47.8% (+5.68 percentage points), even when sub-optimal hyperparameters are used. These findings also extend to the non-linear evaluation setting, (see the lower half of Tab. 1).

These empirical findings lead to several conclusions. First, we observe that the recommended masking of 75% of image patches is largely sub-optimal *across* datasets. Figs. 5 and 10 report an ablation study of the masking ratios for MAE and PMAE. Fig. 5 (top) shows that, across all five datasets, a 75% masking ratio is sub-optimal. For PMAE, the masking ratio seems to be a more stable hyperparameter. Fig. 5 (bottom) shows that across all evaluated datasets we observe the best or near-optimal performance for PMAE at 10% to 20% of the variance masked. Second, we validate the empirical benefits brought by PMAE. Interestingly, we notice that PMAE without any hyperparameter tuning ($PMAE_{rd}$) outperforms or performs similarly to MAE with optimum masking ratio in all but one case (i.e., TinyImageNet with MLP probing). Finally, investing in hyperparameter tuning for PMAE leads to substantial performance gains over MAE.

Fig. 6 provides a deeper look into downstream performance across different training epochs and the variability of results with varying masking ratios. As shown in Fig. 6 (left), the advantages of PMAE become evident after just a few hundred training epochs. Notably, training PMAE for 200 epochs exceeds the performance of MAE after 800 epochs. Additional figures for other datasets can be found in Appx. A.7. Furthermore, Fig. 6 (right) explores how downstream accuracy fluctuates with different masking ratios. Our analysis reveals that PMAE displays comparable or lower standard errors across these conditions in contrast to MAE. Collectively, these results suggest that PMAE's masking strategy enhances the alignment between image reconstruction and image classification tasks more effectively than the MAE objective.

# 6 UNDERSTANDING PMAE

In this section, we aim to provide more intuition as to why masking *components* rather than *image patches* leads to more robust objectives. In Section 2, we discuss the hypothesis under which MIM operates (Kong et al., 2023) and present an example failure case of spatial masking in Fig. 2. We highlight how masking image patches can lead to a misalignment between the MIM objective and the learning of meaningful representations. If all patches covering an object are masked out, it is uncertain whether the remaining patches share any information with the object. Contrary, if the masked out information is redundant with the information carried by visible patches, it is likely that the information shared does not contain the object class but rather perceptual features (e.g., colors or textures).

Different from spatial masking, masking principal *components* leads to the removal of global image features, instead of only acting locally as in spatial masking. Fig. 7 serves as an example

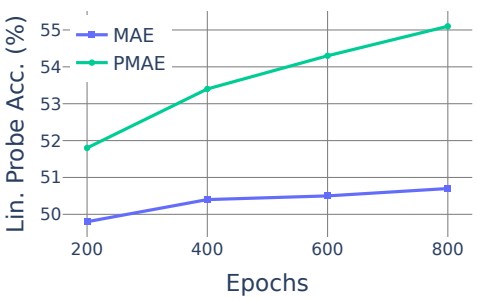 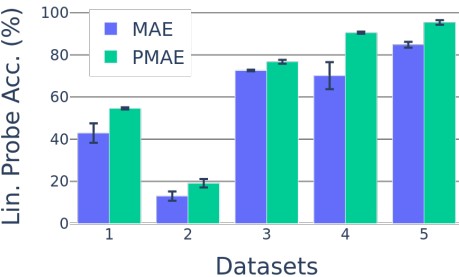

Figure 6: **(Left) Learning curves.** Linear probe accuracy for CIFAR10 classification across the number of training epochs. PMAE outperforms MAE's final performance even after short training times. **(Right) Ablation study of the masking ratio.** Average and standard error of the linear probe accuracy across masking ratios for MAE and PMAE. We observe lower or equivalent standard errors for PMAE than for MAE across CIFAR10 (1), TinyImageNet (2) and MedMNIST (3-5) datasets.

highlighting the correspondence between principal component and perceptual features. In this example, the principal components with the highest eigenvalues capture the colors within the image while the bottom PCs highlight the edges. Early work in image processing (Turk & Pentland, 1991) has demonstrated this connection between an image's dominant modes of variation and its low spatial frequency components, providing further intuition for how information is partitioned in the space of PCs of natural images.

By removing a subset of principal components, PMAE prevents the removal of all the information characterizing an object and prevents redundant information to remain after masking. Instead, PMAE drops a set of unique image components. By taking advantage of the information partitioning in PCA, PMAE thereby mitigates MAE's failure cases, ultimately leading to increased accuracy. Although the potential of the principal component space for Masked Image Modelling (Balestriero & LeCun, 2024) or Image Denoising (Chen et al., 2024b) has been recently explored, our work is the first to propose an effective masking strategy that directly leverages PCA.

## 7 RELATED WORK

**Self-supervised learning.** Self-supervised learning (SSL) leverages auxiliary tasks to learn from unlabeled data, often outperforming supervised methods on downstream tasks. SSL can be divided into two categories: discriminative and generative (Liu et al., 2021). Discriminative methods (Chen et al., 2020a; Caron et al., 2021) focus on enforcing invariance or equivariance between data views in the representation space, while generative methods (He et al., 2021; Bizeul et al., 2024) rely on data reconstruction from, often, corrupted observations. Though generative methods historically lagged in performance, recent work has bridged the gap by integrating strengths from both paradigms(Assran et al., 2022; Dong et al., 2023; Oquab et al., 2023; Chen et al., 2024a; Lehner et al., 2023). Interestingly, recent discriminative methods employ multi-cropping strategies to create distinct data views (Oquab et al., 2023; Assran et al., 2023), which is reminiscent of image masking. Balestriero & LeCun (2024) point out the misalignment between auxiliary and downstream tasks in reconstruction-based SSL and suggest novel masking strategies to help realign these objectives.

**Masked Image Modelling.** MIM extends the successful masked language modeling paradigm to vision tasks. Early methods, such as Context Encoder (Pathak et al., 2016), used a convolutional autoencoder to inpaint a central region of the image. The rise of Vision Transformers (ViTs) (Dosovitskiy et al., 2021) has driven significant advancements in MIM. BEiT (Zhou et al., 2021; Bao et al., 2022) combines a ViT encoder with image tokenizers (Ramesh et al., 2021) to predict discrete tokens for masked patches. SimMIM (Xie et al., 2022) simplifies the task by pairing a ViT encoder with a regression head to directly predict raw pixel values for the masked regions. MAE (He et al., 2021) introduces a more efficient encoder-decoder architecture, with a shallow decoder. MIM's domain-agnostic masking strategies have also proven effective in multi-modal tasks (Baevski et al., 2022; Bachmann et al., 2022).

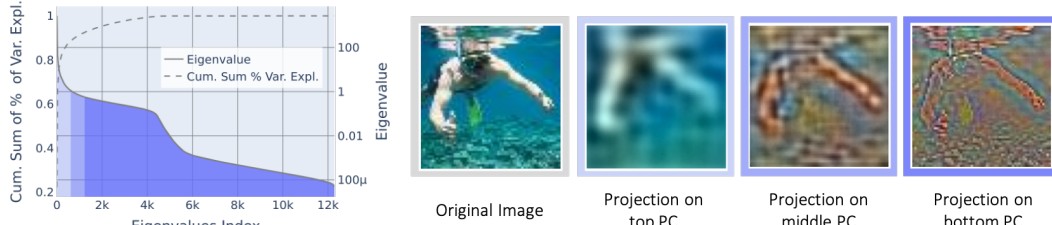

Figure 7: **From Principal Components to Spatial Features.** Overview of the spatial features associated with distinct regions of the principal component (PC) spectrum; Images depict the features encapsulated by the top PCs (light blue), middle PCs (mild blue) and bottom PCs (dark blue).

**Mask Design Strategies.** A critical component of the Masked Image Modeling paradigm is the design of effective masking strategies. Early MIM approaches have relied on random spatial masking techniques, such as masking out the central region of an image (Pathak et al., 2016), image patches (He et al., 2021; Xie et al., 2022), and blocks of patches (Bao et al., 2022). Inspired by advances in language modeling, recent efforts have explored semantically guided mask design. Li et al. (2021) use self-attention maps to mask irrelevant regions, while Kakogeorgiou et al. (2022) focus on masking semantically rich areas. Shi et al. (2022) design masks through adversarial learning, where the resulting masks resemble semantic maps, a concept extended by Li et al. (2022a) through progressive semantic region masking. Further advancing this direction, Wang et al. (2023) and Madan et al. (2024) introduce curriculum learning-inspired mask design methods. These methods often require additional training steps, components, or more complex objectives. More closely related to our work, Chang et al. (2022); Chen et al. (2024b) explore the use of pre-existing image representations for Masked Image Modeling and image denoising. Chen et al. (2024b) introduce additive Gaussian noise to principal components as an alternative to the traditional Denoising Autoencoders. Chang et al. (2022) utilize masked token modeling by leveraging the discrete latent space of a pre-trained VQVAE to develop an image generation model.

## 8 DISCUSSION

In this work, we have investigated different masking strategies for Masked Image Modelling (MIM). To this end, we have introduced the Principled Masked Autoencoder (PMAE) as an alternative to masking random patches of pixels. PMAE is rooted in Principal Component Analysis (PCA; Pearson, 1901; Hotelling, 1933), which is a widely used data-driven *linear* transformation. Unlike recent alternatives that require additional supervision, learnable components, or complex training pipelines (Li et al., 2021; 2022a; Kakogeorgiou et al., 2022; Li et al., 2022b), PMAE stays close to the core principles of MIM: the combination of a randomized masking strategy and an encoder-decoder architecture. Despite its simplicity, we demonstrate that PMAE yields substantial performance improvements over spatial masking on image classification tasks. Further, in a PMAE, the masking ratio—typically a sensitive and difficult-to-tune hyperparameter in MIM—appears more robust and has a natural interpretation as the ratio of variance explained by the masked input.

Since PCA is easily applicable to any data modality, our proposal of masking principal components is not specific to MIM. Instead, it can be viewed as a *general strategy* that should also be applicable to other types of modalities beyond images, as well as to other self-supervised learning (SSL) approaches. Indeed, data masking is commonly adopted in discriminative SSL methods. Whereas early approaches, such as SimCLR (Chen et al., 2020a) or MoCo (Chen et al., 2020b), relied on combinations of image transformations (e.g., color jitter, flips, crops, etc.) as data augmentation strategies, more recent state-of-the-art methods like DINO (Caron et al., 2021; Oquab et al., 2023) and I-JEPA (Assran et al., 2023) have shifted to relying solely on image cropping, which can be considered a type of masking. The integration of principal component masking instead of image cropping into such SSL pipelines constitutes a promising future direction of research.

In the present work, we have focused on PCA as a meaningful masking space. However, our core idea of masking a *transformed* version of the data (rather than the *raw* data) can be viewed as laying the groundwork for other, more generic approaches to information masking in self-supervised

representation learning. Moving beyond PCA, a natural extension would be to *learn* a suitable latent space in which the masking is performed. This route has the added potential of leveraging recent theoretical insights (Kong et al., 2023) by more explicitly enforcing that the shared information between visible and masked-out latent components contains high-level latent variables that are most useful for the downstream tasks of interest. Other off-the-shelf *non-linear* transformations, such as the Fourier transform (Bracewell & Kahn, 1966), Wavelet transform (Daubechies, 1992), Kernel Principal Component Analysis (Schölkopf et al., 1997), or Diffusion Maps (Coifman & Lafon, 2006), represent alternative candidate transformations. Future research should explore whether the properties of these spaces provide comparable or additional advantages over PCA. Preliminary results on Kernel PCA, presented in Appx. A.7.4, demonstrate performance gains over PMAE, motivating further exploration. A particularly appealing aspect of some of these methods (e.g., Fourier & Wavelet transforms and Diffusion Maps) is the use of fixed bases, which could eliminate the computational overhead of PCA—whose cost scales cubically with the data dimensionality—and improve scalability to larger datasets.

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

# A    APPENDIX

## A.1    PRINCIPAL COMPONENT ANALYSIS

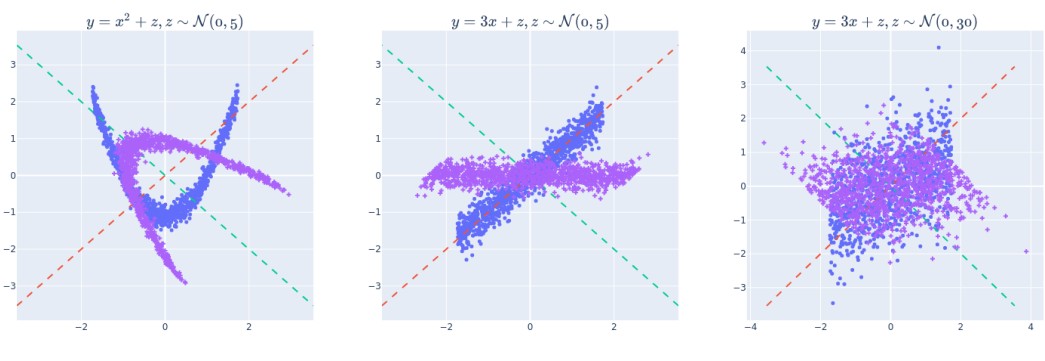

Figure 8: **PCA and independent sources:** PCA finds an orthonormal basis — a rotation matrix — that maximizes the variance in the data. When the original data is a linear combination of independent components (*middle & right*), referred to as data sources, PCA *might* successfully identify these sources, resulting in statistically independent variables when the data is projected onto its principal components. However, if the original data is a non-linear combination of independent sources (*left*), the projection onto the principal components results in statistically dependent variables making it possible to *approximately* predict one PC from others.

Principal Component Analysis (PCA; Pearson, 1901; Hotelling, 1933) finds a set of orthonormal vectors ($\mathbf{V}$) that maximize the variance of the data. Projecting data samples $\mathbf{X}$ onto $\mathbf{V}$ effectively rotates the original data such that each variable captures the maximum possible variance while remaining orthogonal to the previous dimensions. When the original data is a linear combination of independent factors (referred to as sources), PCA *can* but does not necessarily recover these sources. PCA recovers mutually statistically independent sources if and only if the original data is jointly Gaussian, since uncorrelatedness implies independence only for the Gaussian distribution. This situation arises, e.g., when the data is a linear combination of Gaussian sources.

In Fig. 8 (*middle & right*), the observed variables $x$ and $y$ are linear combinations ($y = 3x + z$) of independent sources $x$ and $z$, where $x$ follows a uniform distribution and $z$ is drawn from a normal distribution. In the middle example, PCA finds principal axes, PC1 and PC2, which are orthogonal, and the projections align with the independent sources $x$ and $z$. In the right example, the projections do not align with the independent sources $x$ and $z$.

However, as shown in Fig. 8 (*left*), when $y$ is a non-linear combination of sources ($y = x^2 + z$), no rotation matrix can transform the data to recover statistically independent variables. Projecting the data onto principal axis will hence result in statistically dependent variables making it possible to approximately predict one PC from others.

In this work, we incorporate PCA into the framework of Masked Image Modeling (MIM). Specifically, we propose a task that involves reconstructing masked principal components using the visible ones. This task is meaningful only if it is feasible to approximate the masked PCs based on the visible ones, which occurs when the masked PCs are statistically dependent on the visible PCs.

## A.2    EXPERIMENTAL SETUP

### A.2.1    DATASETS

**CIFAR-10**    is a widely used benchmark dataset containing 50,000 training and 10,000 validation 32x32 RGB images depicting 10 object classes, such as airplanes, cars, and animals.

**TinyImageNet** is a smaller subset of the ImageNet dataset, containing 200 classes of 64x64 RGB images. It consists of 100,000 training images and 10,000 validation images, making it a challenging benchmark for classification tasks with more fine-grained object categories compared to CIFAR-10.

The MedMNIST (Yang et al., 2023) datasets are a collection of medical imaging datasets, each focusing on different types of biomedical data. Three subsets from MedMNIST are used:

**BloodMNIST** consists of 12,000 training and 1,700 validation 64x64 RGB images across 8 classes and represents microscopic images of blood cells, making it useful for classification tasks in hematology.

**DermaMNIST** contains 7,000 training and 1,000 validation 64x64 RGB images across 7 classes and depicts dermatological images of various skin conditions, serving as a tool for diagnosing skin diseases.

**PathMNIST** comprises 90,000 training and 10,000 validation 64x64 RGB images across 9 classes and depicts histopathological images of colorectal cancer tissue, aiding in classification tasks relevant to pathology.

We apply an equivalent data augmentation strategy to all datasets and for all learning objectives during training; Following He et al. (2021), our augmentation strategy consists of a random cropping followed by image resizing using bicubic interpolation. The scale of the random cropping is fixed to $[0.2, 1.0]$. We add horizontal flipping and we normalize images using each dataset's training mean and standard deviation; For evaluation, we resort to image normalization only. For all datasets and methods we define image patches as patches of 8x8 pixels.

### A.3 MODEL ARCHITECTURE

We train a tiny Vision Transformer encoder architecture (ViT-T) with image patch size 8x8 (ViT-T/8). The specifics of this architecture can be found in Tab. 2.

| config | value |
| --- | --- |
| hidden size | 192 |
| number of attention heads | 3 |
| intermediate size | 768 |
| norm pixel loss | True |
| patch size | 8x8 |

Table 2: Model architecture hyperparameters ViT-T/8.

### A.4 TRAINING HYPERPARAMETERS

We train the ViT-T encoder-decoder architecture for 800 epochs with the hyperparameters found in Tab. 3. These hyperparameters are taken from (He et al., 2021). We use the linear lr scaling rule: lr = base lr×batchsize / 256 (Goyal, 2017). Note that for our oracle masking setting, we conduct an ablation study across a masking ratio range of $[0.1, 0.9]$.

### A.5 EVALUATION HYPERPARAMETERS

We evaluate the learned representation (i.e., `[CLS]` token) using a linear probe, multi-layer perceptron classifier and $k$-Nearest Neighbors algorithm on top of the frozen representation. The training samples of each dataset are used to training and the validation samples for testing. For linear and mlp probing experiments, we train the probes for 100 epochs following common practices (He et al., 2021). For the $k$-NN algorithm we tune the number of neighbors in the range $[2, 20]$. More details regarding the linear probing evaluation hyperparameters can be found in Tab. 4. We use the linear

| config | value | | config | value |
|---|---|---|---|---|
| batch size | 512 | | batch size | 512 |
| base learning rate | 0.00015 | | base learning rate | 0.1 |
| optimizer | AdamW [39] | | optimizer | SGD [6] |
| betas (AdamW) | $\beta_1, \beta_2 = 0.9, 0.95$ | | betas (SGD) | 0.9 |
| weight decay | 0.05 | | learning rate | 0.2 |
| learning rate (warmup) | 0.0003 | | warmup steps | 10 |
| warmup steps | 40 | | weight decay | 0 |

Table 3: Training hyperparameters.  Table 4: Linear probing hyperparameters.

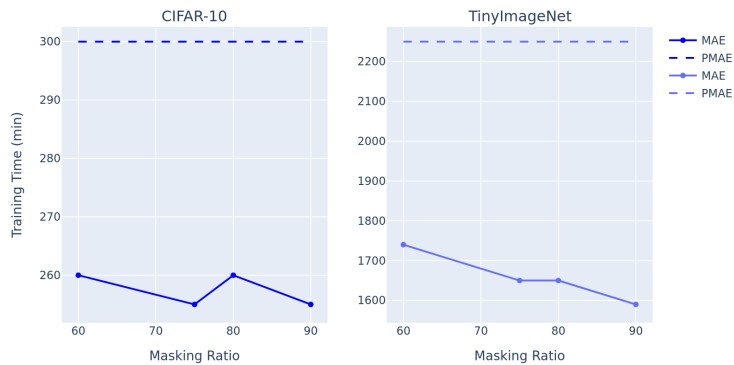

Figure 9: **Training time.** We report the raining time in minutes for 800 training epochs using a ViT-T/8 architecture. For standard MAE we report numbers for various masking ratios.

lr scaling rule: lr = base lr×batchsize / 256 (Goyal, 2017). Note that for PMAE, we evaluate the approach on raw images and do not perform any filtering of principal components prior to evaluation.

## A.6 COMPUTATIONAL RESOURCES

All training runs were conducted on single NVIDIA GeForce RTX 3090/NVIDIA GeForce RTX 4090/Quadro RTX 6000 GPUs or NVIDIA TITAN RTX each of which possesses a 24GB RAM. Fig. 9 reports the time taken for 800 training epochs using a ViT-T/8 architecture on a Quadro RTX 6000 GPU for CIFAR10 and TinyImageNet with the standard MAE and the PMAE methods.

## A.7 ADDITIONAL RESULTS

### A.7.1 MASKING RATIO ABLATION

In Fig. 5b we report the image classification accuracy with a linear probe for our PMAE for different masking ratios in the $[10, 50]$ range. In Fig. 10 we extend this range for completeness to $[10, 90]$. Conclusions drawn from Fig. 5b remain: the optimal masking ratio across datasets lies between 10 and 20% of the variance masked. Above these ratios, we observe a performance drop across datasets.

### A.7.2 TRAINING DYNAMICS

Tab. 5 shows the classification accuracy for CIFAR10, TinyImageNet, BloodMNIST, DermaMNIST and PathMNIST datasets using a $k$-NN classifier in place of a linear probe or MLP probe as presented in Section 5.

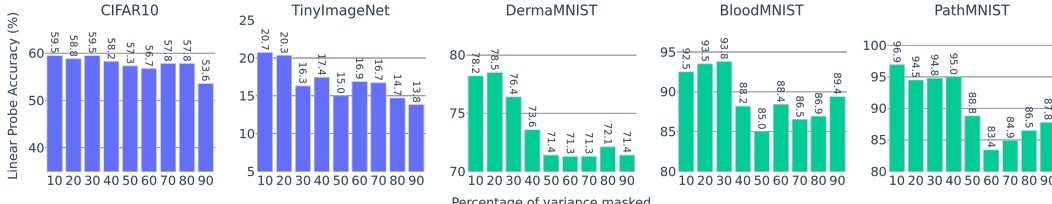

Figure 10: **Impact of the Masking Ratio.** PMAE linear probing accuracy for varying masking ratios. We observe a close to optimal performance across datasets for 10 to 20% of the data variance masked.

Fig. 11 displays the linear probe accuracy for varying training epoch checkpoints. Similar to Fig. 11b we observe that PMAE after 200 epochs outperforms MAE after 800 epochs. For TinyImageNet, PMAE after 200 epochs performs near MAE after 800 training epochs.

Table 5: $k$-Nearest Neighbors top-1% accuracy for CIFAR10, TinyImageNet, DermaMNIST, Blood-MNIST, and PathMNIST for random masking in pixel (*MAE*) and principal component (*PMAE*) space with the standard 75% masking ratio (*std*), oracles (*ocl*) and randomized (*rd*) masking ratios. We report the accuracy after 800 epochs of training using a ViT-T/8 is reported.

|  |  | CIFAR10 | TinyImageNet | DermaMNIST | BloodMNIST | PathMNIST |
|---|---|---|---|---|---|---|
| | $\text{MAE}_{\text{std}}$ | 38.3 | 10.0 | 71.1 | 65.7 | 92.1 |
| | $\text{MAE}_{\text{ocl}}$ | 47.6 | **12.5** | 69.9 | 73.6 | 94.6 |
| $k$-**NN** | $\text{PMAE}^*_{\text{ocl}}$ | **48.1** | 9.6 | **74.7** | **84.5** | **99.1** |
| | $\text{MAE}_{\text{rd}}$ | 40.3 | 7.6 | **71.6** | **82.7** | **96.0** |
| | $\text{PMAE}^*_{\text{rd}}$ | **49.6** | **9.5** | 70.6 | 76.0 | 94.8 |

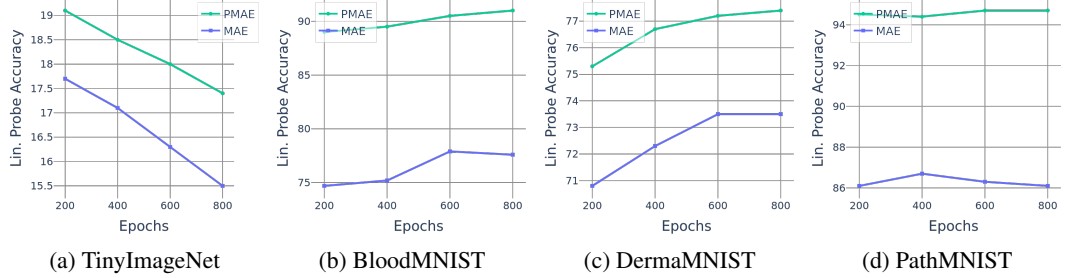

| (a) TinyImageNet | (b) BloodMNIST | (c) DermaMNIST | (d) PathMNIST |

Figure 11: **Performance Curves.** Linear probe accuracy (%) for TinyImageNet, BloodMNIST, DermaMNIST and PathMNIST across training epochs. In MedMNIST datasets we observe that PMAE after 200 epochs outperforms MAE after 800 epochs. For TinyImageNet, PMAE after 200 epochs performs near MAE after 800 training epochs.

### A.7.3 RECONSTRUCTING IN PIXEL VS. PRINCIPAL COMPONENT SPACE

We further investigate the impact of the domain (i.e., pixel vs. pc space) in which the reconstruction error is minimized on downstream performance. In Fig. 12, we present an alternative to Fig. 1 in which the training objective receives a set of principal components in place of pixels. In Fig. 12, the decoder's output is projected onto the data's principal axes. The training objective then minimizes the Euclidean distance between the ground truth and the predicted masked principal components. Instead with Fig. 1, the training objective minimizes the Euclidean distance between the ground truth masked principal components projected back to pixel space and the decoder's output. The learning objective then becomes Eq. (A.1), a modified version of Eq. (3.1):

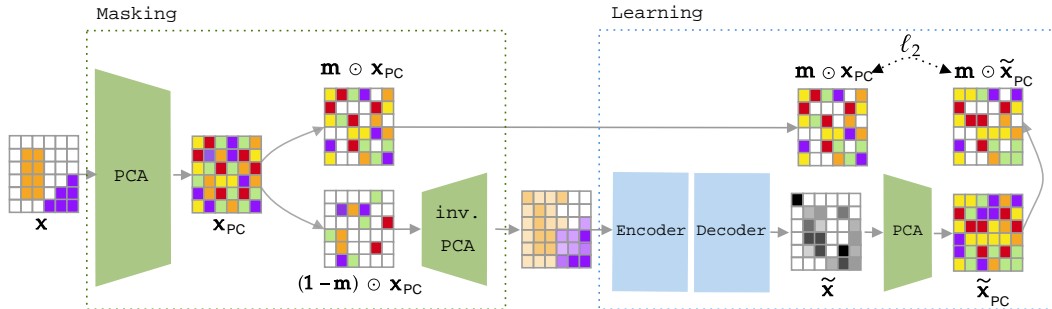

Figure 12: **Overview of the Principled Masked Autoencoder (PMAE) with masked principal components as reconstruction target.** A Principled Masked Autoencoder (PMAE) differs from a vanilla MAE by performing the masking in the space of principal components $\mathbf{x}_{PC} = \text{PCA}(\mathbf{x})$ rather than in the observation space. The visible principal components $(\mathbf{1} - \mathbf{m}) \odot \mathbf{x}_{PC}$ are then projected back into the observation space and serve as the input for an encoder-decoder architecture. Masked principal components, $\mathbf{m} \odot \mathbf{x}_{PC}$, serve as the reconstruction target.

$$\mathcal{L}_{\text{PMAE}}(\mathbf{x}, \mathbf{m}; \boldsymbol{\theta}, \boldsymbol{\phi}) = \| \mathbf{m}^{\mathbf{c}} \odot t(g_{\boldsymbol{\theta}}(f_{\boldsymbol{\phi}}(h(\mathbf{m}, \mathbf{x})))) - \mathbf{m}^{\mathbf{c}} \odot t(\mathbf{x}) \|_2^2 , \tag{A.1}$$

Tab. 7 presents the downstream image classification performance achieved when training representations with Eq. (A.1). In particular, it reports results obtained using a linear and MLP probe in the oracle setting (i.e., with optimal masking ratios). PMAE consistently outperforms MAE across all five datasets, demonstrating substantial improvements. Notably, the performance gains over MAE are larger than those observed for representations trained with Eq. (3.1), reported in Tab. 1. In Tab. 1 we report an average performance gain of 6.6 percentage points across datasets over the MAE baseline, while Tab. 7 reports an average performance gain of 9.6 percentage points. These findings further support our claims that the space of principal components constitutes a meaningful masking space for Masking Image Modelling learning paradigms. Note that the optimal masking ratios used in Tab. 1 for each dataset, are the ones reported in Fig. 5b.

Table 6: Linear and MLP probe top-1% accuracy for CIFAR10, TinyImageNet and MedMNIST datasets for random masking in pixel (MAE) and principal component (PMAE) space with the standard 75% masking ratio (std) and oracles (ocl). The reconstruction target for PMAE lies here in the space of principal components. ⋆ refers to ours.

|  |  | CIFAR10 | TinyImageNet | DermaMNIST | BloodMNIST | PathMNIST |
|---|---|---|---|---|---|---|
| **Linear** | $\text{MAE}_{\text{std}}$ | 41.7 | 11.5 | 72.4 | 73.4 | 83.4 |
|  | $\text{MAE}_{\text{ocl}}$ | 50.7 | 15.5 | 73.7 | 78.6 | 86.4 |
|  | $\text{PMAE}^{\star}_{\text{ocl}}$ | **59.0** | **22.5** | **95.5** | **78.6** | **96.8** |
| **MLP** | $\text{MAE}_{\text{std}}$ | 34.0 | 15.5 | 72.2 | 68.6 | 92.6 |
|  | $\text{MAE}_{\text{ocl}}$ | 55.2 | 22.2 | 74.4 | 75.8 | 95.1 |
|  | $\text{PMAE}^{\star}_{\text{ocl}}$ | **64.1** | **25.1** | **92.5** | **80.2** | **98.6** |

Table 7: Linear and MLP probe top-1% accuracy for CIFAR10, TinyImageNet and MedMNIST datasets for random masking in pixel (MAE) and principal component (PMAE) space with the standard 75% masking ratio (std) and oracles (ocl). The reconstruction target for PMAE lies here in the space of principal components. ⋆ refers to ours.

|  | CIFAR10 | TinyImageNet | DermaMNIST | BloodMNIST | PathMNIST |
|---|---|---|---|---|---|
| $\text{MAE}_{\text{ocl}}$ | 80.5 | 42.8 | 79..9 | 98.1 | 99.7 |
| $\text{PMAE}_{\text{ocl}}$ | **84.8** | **44.5** | **82.3** | 98.1 | 99.7 |

Table 8: Linear and MLP probe top-$1\%$ accuracy for CIFAR10 for random masking in pixel (MAE), in principal component space (PMAE) and in kernelized PCA space (KMAE) with the standard $75\%$ masking ratio (std) and oracles (ocl). The reconstruction targets for PMAE and KMAE lie in the space of principal components. * refers to ours.

|        | $\mathbf{MAE}_{\text{std}}$ | $\mathbf{MAE}_{\text{ocl}}$ | $\mathbf{PMAE}^*_{\text{ocl}}$ | $\mathbf{KMAE}^*_{\text{ocl}}$ |
|--------|------|------|------|------|
| Linear | 41.7 | 50.7 | 59.0 | **64.0** |
| MLP    | 34.0 | 55.2 | 64.1 | **68.6** |

### A.7.4 BEYOND PCA

Our work shows evidence the PCA offers a meaningful masking space. In Section 6, we motivate our choice by observing that principal components capture global rather than local features of an image. In this section, we go beyond PCA and explore non-linear matrix factorization methods as a proof of concept for future research. In particular, we explore kernel PCA (Schölkopf et al., 1997) with a Radial Basis Function. In kernel PCA, the spectral decomposition is performed not on the data itself but rather on a modified version of it: the standardized data is mapped to a high-dimensional space via a non-linear kernel function.

In Tab. 8, we present results on the CIFAR10 dataset and show the image classification accuracy using a linear and MLP probe. We compare a vanilla MAE with our PMAE and KMAE which relies on Kernel PCA for optimal masking ratios. For KMAE, we use the setting presented in Appx. A.7.3 and minimize the Euclidean distance between masked principal components and the decoder's output principal components.

The results reveal a significant performance improvement when employing a non-linear image transformation. KMAE achieves an average gain of 13.3 and 5 percentage points compared to the standard Masked Autoencoder (He et al., 2021) and PMAE, respectively. Although these findings are preliminary and based on a single mid-scale dataset, they highlight the potential of non-linear transformations and further emphasize the value of spectral decomposition as a meaningful for Masked Image Modeling paradigms.

