# OpenReview forum: "Components Beat Patches: Eigenvector Removal for Robust Masked Image Modelling"
_ICLR.cc/2025/Conference — Submitted to ICLR 2025_

### Official Review · Reviewer_1XJX · 2024-10-21

**Soundness:** 2
**Presentation:** 3
**Contribution:** 1
**Rating:** 3
**Confidence:** 4

**Summary:**

The authors present a variant of the masked autoencoder where instead of masking a subset of image patches, they mask a subset of principal components. They assume that the masked and unmasked principal components are likely to be correlated in a way that is pertinent to the class, which improves downstream classification performance.

**Strengths:**

The paper is well-written, and provides a clear and agreeable motivation for masking principal components instead of image patches.

**Weaknesses:**

The main purpose of self-supervised learning is to learn a representation that can be fine-tuned to achieve **strong** downstream performance. This is not demonstrated in this paper. Putting it bluntly, it is not compelling to achieve ~60% accuracy on CIFAR10 and ~20% accuracy on TinyImageNet. Instead of using significant resources to train a vision transformer with this approach, it's more practical to train a small convolutional net to achieve higher performance.

In the original masked autoencoder paper, the authors demonstrate very strong downstream performance on ImageNet1k (not the tiny version), which makes their approach compelling. In light of this, the fact that MAE allegedly performs so poorly on CIFAR10 and TinyImageNet is a bit suspicious. If it is indeed the case that MAE performs poorly on smaller images, we still need to see PMAE do similarly well on ImageNet1k, but these experiments are not in the paper.

**Questions:**

Can you explain why it makes sense for MAE to achieve near-SOTA performance on ImageNet1k in the original paper, but performs so poorly on the simpler datasets CIFAR10 and TinyImageNet?

As stated in "weaknesses," I think we need to see PMAE's performance on ImageNet1k to reach a conclusion about it's superiority over MAE.

---

> ### Author Response · Authors · 2024-11-22
> **Rebuttal Part 1**
>
> Dear reviewer 1XJX,
>
> Thank you for your review of our work. Please find below our answers to your questions:
>
> #### **Baseline performance**
>
> Our work proposes the masking of principal components rather than pixels. We view this contribution as a proof-of-concept, demonstrating that masking in the space of principal components can significantly enhance downstream performance and inspire further research in Masked Image Modeling. This perspective aligns with feedback from other reviewers, such as reviewer WJuv, who noted: _"The idea is simple yet solid, even drawing intuitions from early research in image processing / Eigenfaces, which I can imagine inspiring future/new directions in representation learning”_. With this goal in mind, we verify our claims by conducting experiments on a **ViT-tiny for 800 training epochs** on 5 datasets spanning across different domains (i.e., natural and medical images).
>
> A ViT-tiny is a smaller-scale architecture of roughly 30.8M parameters (~3x and ~10x less than a Vit-B and Vit-L on which ImageNet1k is commonly evaluated). It is often used for datasets such as CIFAR10 and TinyImageNet. The following pointers (recently published) provide insights into the standard range of linear probing performance achieved for CIFAR10 and TinyImageNet datasets with MAEs with such architecture.
>
> Table 1 in [2] reports a performance of 59.6% for CIFAR10 after 2000 training epochs with a Vit-tiny. Table 2 in [3] trains a Vit-tiny on CIFAR10 and TinyImageNet for 2000 and 1000 epochs, respectively, and shows a classification performance with a linear probe of 72.5% and 19.6%, respectively.
>
> Beyond the architecture scale, fine-tuning is expected to yield significantly higher performance compared to using a linear probe. Table 2 of [4] presents fine-tuning results on a ViT-B architecture for CIFAR-10 and TinyImageNet, achieving image classification accuracies exceeding 90% and 60%, respectively. These observations collectively indicate that lower linear probing scores on a ViT-tiny are anticipated and should not be interpreted as poor performance of MAE on smaller images.
>
> Additionally, we ensured that only minimal modifications were made to the original MAE codebase [1], maintaining consistency between the original approach and our reported results. Our codebase is included in the supplementary materials submitted with the manuscript.
> We hope these explanations clarify why the baseline scores (MAE) reported in our work are lower than the current state-of-the-art for these datasets. As highlighted by other reviewers --- _“The evaluation strategy is solid and the use of SOTA baselines is good.”_ (reviewer 5bnU) ---, we believe our empirical validation provides strong evidence that the space of principal components is a meaningful alternative to the image space for self-supervised representation learning.
>
> We welcome any additional questions you may have regarding our implementation of the baselines.
>
> [1] https://github.com/facebookresearch/mae
>
> [2] Zhang, Qi, Yifei Wang, and Yisen Wang. "How mask matters: Towards theoretical understandings of masked autoencoders." Advances in Neural Information Processing Systems 35 (2022): 27127-27139.
>
> [3] Zhang, Kevin, and Zhiqiang Shen. "i-mae: Are latent representations in masked autoencoders linearly separable?." Proceedings of the IEEE/CVF Conference on Computer Vision and Pattern Recognition. 2024.
>
> [4] Mao, Jiawei, et al. "Masked autoencoders are effective solution to transformer data-hungry." arXiv preprint arXiv:2212.05677 (2022).

---

> ### Author Response · Authors · 2024-11-22
> **Rebuttal Part 2**
>
> #### **Scalability of PMAE to large datasets**
>
> As PMAE relies on PCA, extending our experimental setup to larger-scale datasets would require the computation of their principal components. As rightfully mentioned by reviewer WJuv, the cost of this operation scales cubically with the data dimensionality (i.e., $O(d^2n+d^3)$ for a dataset with $n$ samples of dimensionality $d$). To lower this cost, especially for larger images like ImageNet1k, future work could consider other more cost-effective data transformations (please see the general answer to reviewers for a more thorough discussion on future directions for scalability).
>
> While these additional costs currently limit the applicability of PMAE to larger datasets, our work shows substantial performance gains obtained with PCA on a range of medium-sized datasets. These findings lay the groundwork for the exploration of other image transformations, perhaps more cost-effective, as mappings to a latent space in which masking is performed. Besides, we do not intend to claim that PMAE is superior to MAE in all scenarios, including on large-scale and very high-dimensional datasets. Rather we view our work as a proof of concept that masking in a learnt latent space can be beneficial for self-supervised masked image modelling. We provide clear evidence of this for several medium-sized settings, but agree with the reviewer that further work (e.g., transformations beyond standard PCA) is needed to scale our alternative masking strategy to more complex scenarios.
>
> Thanks to your feedback, we updated section 8 of the manuscript to further discuss the scalability of our approach as follows: _“Other off-the-shelf non-linear transformations, such as the Fourier transform, Wavelet transform, Kernel Principal Component Analysis, or Diffusion Maps, represent alternative candidate transformations. Future research should explore whether the properties of these spaces provide comparable or additional advantages over PCA. [...] A particularly appealing aspect of some of these methods (e.g., Fourier \& Wavelet transforms and Diffusion Maps) is the use of fixed bases, which could eliminate the computational overhead of PCA---whose cost scales cubically with the data dimensionality--- and improve scalability to larger datasets.”_
>
> Thank you again for your feedback. We welcome any additional suggestions, questions, or requests for information and encourage further discussions.
>
> Finally, please refer to the general answer to reviewers for additional results added during this rebuttal that further support our claims.

---

> > ### Comment · Reviewer_1XJX · 2024-11-25
> >
> > Thank you for your response.
> >
> > The fact that this algorithm doesn't scale to ImageNet-sized images (which are not large by today's standards) is problematic. This means it needs to really perform on smaller images, which it doesn't. To reiterate my question, why would I want to train a 30M parameter vision transformer for 2000 epochs to obtain 60% accuracy on CIFAR10?

---

> > > ### Author Response · Authors · 2024-11-26
> > >
> > > Dear reviewer 1XJX,
> > >
> > > We greatly appreciate your engagement in the discussion.
> > >
> > > Firstly, we would like to clarify that the near-SOTA performance of MAE on ImageNet mentioned in your review pertains to the fine-tuning setting, not to evaluation using a linear probe. It is well established that MAE models perform particularly well in fine-tuning scenarios.  Below we provide additional results when fine-tuning the encoder rather than simply training a linear probe for MAE and PMAE for optimal masking ratios:
> > >
> > > | Dataset      | MAE  | PMAE  |
> > > |--------------|------------------|--------------------|
> > > | CIFAR-10     | 80.5             | **84.8**          |
> > > | TinyImageNet | 42.8             | **44.5**          |
> > > | BloodMNIST   | 98.1             | 98.1              |
> > > | DermaMNIST   | 79.9             | **82.3**          |
> > > | PathMNIST    | 99.7         | 99.7              |
> > >
> > > **Table 1:** Fine-tuning accuracy for MAE and PMAE across CIFAR-10, TinyImageNet, BloodMNIST, DermaMNIST, and PathMNIST for optimal masking ratios. Bold numbers indicate the best scores.
> > >
> > > Please note that these results were obtained using the fine-tuning parameters in Table 9 of the original MAE paper.
> > >
> > > Now, to address your question: "Why would I want to train a 30M parameter Vision Transformer for 2000 epochs to obtain 60% accuracy on CIFAR-10?" Based on the fine-tuning results provided above, if the goal is absolute performance, training a 30M parameter Vision Transformer with MAE for 800 epochs (+100 epochs of fine-tuning) yields roughly 80% accuracy on CIFAR-10. With PMAE, this increases further to nearly 85%.
> > >
> > > We would also like to explain the relevance of demonstrating performance improvements using PMAE on small to medium-sized datasets. While large datasets (in both resolution and sample size) have become popular for pre-training, they are predominantly composed of natural images and text. These representations may not transfer effectively to other data domains (e.g., medical images) or modalities (e.g., time series, tabular data). Smaller datasets that are closer in nature to the target domain can often yield better transfer performance.
> > >
> > > To illustrate this, we evaluate the transferability of representations learned with medium-sized datasets (e.g., TinyImageNet, PathMNIST) on smaller medical datasets (e.g., BloodMNIST, DermaMNIST). Despite TinyImageNet having more samples than PathMNIST (100k vs. 90k), representations learned from PathMNIST, closer in essence to BloodMNIST and DermaMNIST (all three are medical imaging datasets),  led to improved downstream performance:
> > >
> > > | Pre-training Dataset | BloodMNIST| DermaMNIST|
> > > |-----------------------|-------------------------|--------------------------|
> > > | TinyImageNet          | 80.7               | 73.9                |
> > > | PathMNIST             | **85.8**                   | **76.2**                    |
> > >
> > > **Table 2:** Transfer learning accuracy. Representations are pre-trained on TinyImageNet or PathMNIST, and a linear probe is subsequently trained on frozen representations with the BloodMNIST and DermaMNIST datasets.
> > >
> > > This proof-of-concept highlights that more is not necessarily better depending on the downstream domain of interest. Pre-training on small to medium-sized unlabelled datasets can lead to better downstream performance than pre-training on larger datasets. We hope this proof-of-concept further clarifies why we believe PMAE remains a valuable approach despite current limitations.
> > >
> > > Finally, while we agree that further efforts are needed to address scalability issues, our work demonstrates substantial performance improvements with limited hyperparameter tuning needed. These findings open exciting new avenues for self-supervised learning by challenging the prevailing assumption that the space of observations is optimal for masked image modeling. Our contributions provide meaningful insights to the community and encourage further exploration in this direction.
> > >
> > > We hope these clarifications have helped address your concerns and we welcome any follow-up discussion

---

> > > > ### Author Response · Authors · 2024-12-02
> > > >
> > > > Dear Reviewer 1XJX,
> > > >
> > > > Thank you again for your engagement in the discussion and for the time spent reviewing our work.
> > > >
> > > > As the discussion period is slowly coming to an end, we would greatly appreciate any feedback on our last rebuttal answer which provides additional empirical evidence to address your concerns. If these additional insights helped resolve your concerns, we hope this prompts a reevaluation of our work.

---

### Official Review · Reviewer_5bnU · 2024-11-01

**Soundness:** 3
**Presentation:** 4
**Contribution:** 3
**Rating:** 8
**Confidence:** 4

**Summary:**

The paper proposes a new approach to pre-training neural networks in the context of masked image modelling. Instead of masking out random patches in the image, the authors propose a masking strategy in which the principal components of the image are masked out and the networks are then trained to recover these principal components. The paper empirically demonstrates the advantage of this approach on natural and medical images in the context of image classification.

**Strengths:**

- The idea for the masked image modelling strategy is simple and intuitive. It is presented in an easy-to-follow fashion.
- The paper is generally well-written and clear (except for the points below).
- The evaluation strategy is solid.
- The use of SOTA baselines is good.

**Weaknesses:**

- The idea of using PCA as an invertible transformation into a latent space is sensible, however there many alternative choices for such a transformation (e.g. Fourier or wavelet transforms).
- The evaluation scenario focuses on classification as a downstream task. This is typically a task that relies on global information. However, other downstream tasks, such as object detection or semantic segmentation, require local information, and here, the proposed may perform less well. However, this is not tested.
- The datasets used for evaluation contain very small images (e.g. CIFAR10, MedMNIST)

**Questions:**

- The method uses a lossless scenario for the PCA. The rationale for this is understandable, but I am wondering if this leads to practical problems (e.g. for very small eigenvalues, their ordering becomes random, etc.)?
- Why not evaluate other similar transforms (e.g. Fourier or wavelet transforms)?
- How would the pre-training perform on large, high-resolution images?

---

> ### Author Response · Authors · 2024-11-22
> **Rebuttal**
>
> Dear reviewer 5bnU,
>
> Thank you for your thorough review of our work! Please find below our answers to your questions:
>
> #### **Going beyond PCA**
>
> We agree that many other invertible transformations, including non-linear transformations, could serve as interesting alternatives to PCA. We see our work as proof of concept, showing that masking in latent space instead of the observation space consistently leads to substantial performance gains across datasets. Building on these results, exploring other data transformations, such as the Fourier or wavelet transform, represents promising directions for future research with the potential for additional benefits on downstream tasks.
>
> Following a suggestion from reviewer WJuv, we have conducted preliminary research using Kernel PCA in place of PCA on the CIFAR10 dataset. These additional results were incorporated into the appendix in section A.6.4.
>
> Kernel PCA [1] is a non-linear data transformation, which differs from PCA by performing the spectral decomposition in a latent space in place of the space of observations. The data is first mapped to a high-dimensional space using a kernel function — we chose the Radial Basis Function (RBF) kernel — before spectral decomposition is performed.
>
> Table 1 below reports the image classification accuracy using a linear and MLP probe for CIFAR10 for a standard MAE, PMAE, and KMAE, which relies on Kernel PCA. Results show performance gains of 13.3 and 5 percentage points brought by KMAE over MAE and PMAE, respectively, when using a linear probe. These preliminary findings suggest that masking in latent space can be extended beyond PCA and lead to further empirical gains.
>
> Please note that scores for PMAE reported in table 1 are computed following the alternative objective described in “Additional results supporting our claims” in the general answer to reviewers.
>
> |           | MAE   | PMAE* | KMAE* |
> |-----------|-------|-------|-------|
> | **Linear** | 50.7 | *59.0* | **64.0**  |
> | **MLP**    | 55.2  | *64.1* | **68.6** |
>
> **Table 1**: Linear and MLP probe accuracy for MAE, PMAE, KMAE with best masking ratios for each method. Bold numbers refer to the best scores, italic to the second best. * refers to our work.
>
> Based on your feedback and these interesting additional results, we have improved our discussion in section 8 as follows: _“Other off-the-shelf non-linear transformations, such as the Fourier transform, Wavelet transform, Kernel Principal Component Analysis, or Diffusion Maps, represent alternative candidate transformations. Future research should explore whether the properties of these spaces provide comparable or additional advantages over PCA. Preliminary results on Kernelized PCA, presented in appendix A.6.4, demonstrate performance gains over PMAE, motivating further exploration.”_
>
> [1] Schölkopf, Bernhard, Alexander Smola, and Klaus-Robert Müller. "Kernel principal component analysis." International conference on artificial neural networks. Berlin, Heidelberg: Springer Berlin Heidelberg, 1997.

---

> > ### Author Response · Authors · 2024-11-22
> > **Rebuttal Part 2**
> >
> > #### **PMAE on dense prediction tasks**
> >
> > Although PMAE already demonstrates a clear advantage in standard image classification, we agree with the reviewer that extending our evaluation to include dense prediction tasks would enhance our understanding of PMAE. We are actively working to generate these results and will include these findings in our manuscript once finalized. We appreciate your suggestions, which help improve our work.
> >
> > #### **Practical issues with PCA**
> >
> > The masking strategy proposed in our work, illustrated in Figure 4 of the manuscript, involves randomly selecting a new subset of principal components (PC) to mask at each training step. Although the specific set of masked PCs changes with each step, the total variance explained by the masked PCs remains constant in our oracle approach. Consequently, the exact ordering of PCs is not critical; rather, what matters is the magnitude of their associated eigenvalues. Therefore, we do not anticipate any issues arising from the incorrect ordering of PCs with low eigenvalues.
> >
> > We apologize for any confusion and would be happy to discuss further if we have misunderstood your concern.

---

> > > ### Author Response · Authors · 2024-11-22
> > > **Rebuttal Part 3**
> > >
> > > #### **Very small images vs. Larger datasets**
> > >
> > > Our work proposes the masking of principal components rather than pixels. We see this contribution as a proof-of-concept that masking in latent space can lead to substantial improvements in downstream performance. These claims are supported by empirical evidence across 5 datasets, including three datasets from the MedMNIST dataset.
> > >
> > > The MedMNIST dataset [1] is available in multiple image resolutions. For all three MedMNIST datasets used in our experimental pipeline, we selected the 64x64 image resolution. As a result, our evaluation pipeline includes five datasets, four of which have a 64x64 image resolution (TinyImageNet & MedMNIST). We apologize for any confusion and have clarified this point in section 5.
> > >
> > > As PMAE relies on PCA, extending our experimental setup to larger-scale datasets would require the computation of their principal components. As rightfully mentioned by reviewer WJuv, the cost of this operation scales cubically with the data dimensionality (i.e., $O(d^2n+d^3)$ for a dataset with $n$ samples of dimensionality $d$).
> > >
> > > To lower this cost, future work could consider more cost-effective data transformations. Notably, the Discrete Cosine Transform (DCT [2]), closely related to the Fourier transform, could be a viable cost-effective alternative to PCA. The DCT can be described as a discrete Fourier transform operating on real numbers only and is an approximation of PCA [3]. The advantage of the DCT lies in its cost. Unlike PCA, it relies on a fixed predefined basis and would thereby lower computational costs and help with scalability.
> > >
> > > While PMAE is currently limited to mid-scale datasets, our work presents substantial performance gains obtained with PCA on various datasets. These findings lay the groundwork for the exploration of other image transformations as mappings to a latent space in which masking is performed;
> > >
> > > Thanks to your feedback, we have updated section 8 of the manuscript to further discuss the scalability of our approach as follows: _“Other off-the-shelf non-linear transformations, [...] represent alternative candidate transformations. [...] A particularly appealing aspect of some of these methods (e.g., Fourier \& Wavelet transforms and Diffusion Maps) is the use of fixed bases, which could eliminate the computational overhead of PCA---whose cost scales cubically with the data dimensionality--- and improve scalability to larger datasets.”_
> > >
> > > Please also see the general answer to reviewers for further discussion on scalability.
> > >
> > > Thank you again for your valuable feedback. We welcome any additional suggestions, questions, or requests for information and encourage further discussions.
> > >
> > > [1] https://github.com/MedMNIST/MedMNIST/blob/8cce68f261f993bd0450edc0200498a0691362c2/README.md?plain=1#L8
> > >
> > > [2] Ahmed, Nasir, T_ Natarajan, and Kamisetty R. Rao. "Discrete cosine transform." IEEE transactions on Computers 100.1 (1974): 90-93.
> > >
> > > [3] Sanchez, Victoria, et al. "Diagonalizing properties of the discrete cosine transforms." IEEE transactions on Signal Processing 43.11 (1995): 2631-2641

---

> > > > ### Comment · Reviewer_5bnU · 2024-11-24
> > > >
> > > > Thank you for your answers to my questions. I remain positive about this paper and will keep my rating.

---

### Official Review · Reviewer_tnEx · 2024-11-04

**Soundness:** 2
**Presentation:** 3
**Contribution:** 1
**Rating:** 3
**Confidence:** 2

**Summary:**

This paper proposed a new masked strategy based on principal component analysis, in order to capture more global information. The auto-encoder is employed to reconstruct the masked-out components. Experiments are conducted on various datasets.

**Strengths:**

This paper is well organized and written. The perspective of masking principle components is interesting.

**Weaknesses:**

It does not make sense to me that we can recover removed principal components from other components. This is because unlike patches, these principal components are independent to each other. It is unreasonable to say we can recover a random variable X from another independent random variable Y. This is my main concern for me to give rejection, but I am open to change my score if I am convinced.

Besides, reporting experimental results on ImageNet would be more convincing.

**Questions:**

None.

---

> ### Author Response · Authors · 2024-11-22
> **Rebuttal**
>
> Dear reviewer tnEx,
>
> Thank you for your review of our work.
>
> We would like to clarify some points regarding the proposed work because your main point of criticism is based on a misunderstanding of a fundamental aspect of our work.
>
> Principal component analysis (PCA) finds components that maximize variance in the projected space, resulting in uncorrelated components (i.e., no linear relationship between principal components). However, this does not imply that the principal components are independent, as uncorrelatedness does not imply independence. In contrast, independent component analysis (ICA), which our work does not rely on, seeks to separate data into statistically independent components. In our opinion, it is reasonable to consider that natural or medical images are produced by a non-linear generative process. Consequently, it is unlikely that all principal components of natural or medical images would be independent.
>
> Our empirical results (c.f., Table 1 in our manuscript)  show performance scores way above those of random predictions, which would not be possible if the input image and reconstruction target were to be independent. Examples of visible and masked principled components, provided in Figure 3 (blue) of the original manuscript, also show visual evidence that disjoint sets of principal components are not independent of one another (i.e., the animal contours are present in both the input and target images in Figure 3 (blue)).
>
> Finally, we would like to conclude by emphasizing that other reviewers have recognized the soundness of our approach: “The idea is simple yet solid, even drawing intuitions from early research in image processing / Eigenfaces, which I can imagine inspiring future/new directions in representation learning.” (reviewer WJuv); “The idea of using PCA as an invertible transformation into a latent space is sensible” (reviewer 5bnU).
>
> Further discussion regarding the scaling of our approach to larger-scale datasets is provided in the general answer to reviewers.
>
> Thank you again for your feedback. We hope we could address your concerns. In light of these clarifications regarding some fundamental aspects of our work, we kindly ask you to reconsider your evaluation.
>
> We welcome any additional suggestions, questions, or requests for information and encourage further discussions.

---

> > ### Comment · Reviewer_tnEx · 2024-11-27
> >
> > Although components are just uncorrelated to each other, it still seems confusing for me to reconstruct masked out components using others. This is because components are orthogonal to each other, it is not possible to infer some components from others using linear approximation. Nonlinear approximation may be possible, but not always; therefore, I think the authors should provide conditions under which the reconstruction is possible.

---

> > > ### Author Response · Authors · 2024-11-28
> > >
> > > Dear Reviewer tnEx,
> > >
> > > Thank you for engaging in the discussion.
> > >
> > > To clarify further and avoid any potential confusion for readers, **we have incorporated your feedback into our manuscript**. Specifically, in Section 2 and the new Appendix A.1, we have elaborated on when Principal Component Analysis (PCA) can lead to independent components.
> > >
> > > Notably, **Figure 8 (left) provides a visual example showing that under the assumption of a non-linear generative process, PCA results in uncorrelated but statistically dependent components. This dependency enables one principal component to be approximately predicted from other principal components.**
> > >
> > > To summarize the clarification added to our manuscript:
> > >
> > > - If two variables are statistically dependent, it is possible to approximate one from the other.
> > > - Assuming the data arises from a generative process where independent sources are mixed, PCA outcomes depend on the nature of the generative process:
> > >   - Gaussian linear generative process: PCA identifies statistically independent components, as uncorrelatedness implies independence for Gaussian distributions.
> > >   - Linear generative process: PCA may or may not identify statistically independent components.
> > >   - Non-linear generative process: PCA identifies statistically dependent components.
> > >
> > > Masked Image Modeling explores the natural image domain, where a non-linear generative process is widely considered a realistic assumption.
> > >
> > > Lastly, we would like to remind the reviewer that vanilla Masked Autoencoders similarly rely on the assumption that masked and visible information (e.g., pixels in the case of MAEs) are statistically dependent, allowing masked pixels to be approximated from visible ones.
> > >
> > > We truly hope that these clarifications, along with the refinements in our manuscript, address your concerns and justify a re-evaluation of our work and your score.

---

### Official Review · Reviewer_WJuv · 2024-11-04

**Soundness:** 4
**Presentation:** 4
**Contribution:** 3
**Rating:** 8
**Confidence:** 4

**Summary:**

This paper addresses the problem of Masked Image Modeling, which classically entails masking out patches of pixels within input images and training a model to reconstruct these missing values based on the visible pixels in a self-supervised fashion. The authors propose a novel alternative to the masking of pixel patches, instead suggesting that the data first be transformed into a latent subspace i.e. projected onto its principal components, and masking operations be done on the component level instead of pixel level. The idea is that because the principal components can represent global correlations, masking individual components still retains information at some level of all pixel locations, as opposed to the masking of entire patches of pixels, which could remove e.g. an entire object. In essence, this allows the model to still be exposed to information from all pixel locations during training, leading to robust representations that are more likely to contain meaningful information needed for downstream tasks, e.g. classification. The approach, termed PMAE, is evaluated against the vanilla Masked Autoencoder in an extensive set of experiments for image classification on multiple natural and medical image sets, showing clear and often significant improvement in nearly all settings.

**Strengths:**

- Originality

The proposed method is a relatively straightforward combination of previously established approaches, but integrated in a novel, clever, and elegant way.


- Quality

The authors provide a thorough review of previous research leading to and motivating this work, as well as a nice discussion of the broader context of related research.

Results support claims made throughout the paper.



- Clarity

The paper is very well written and easy to follow; the foundations are solid and motivation is clear.


- Significance

The proposed method avoids extensive hyperparameter tuning, known to be challenging in established MIM/MAE regimes, as ratio / size of image patches masked strongly influences performance on different  downstream tasks.  Experiments presented using MAEs parameterized with range of masking ratios highlight this -- training models with the standard convention of masking $75$% of image patches often results in poorer classification accuracy, suggesting the necessity of tuning beyond this accepted norm.
In contrast, PMAE doesn't require much hyperparameter tuning and seems to perform quite well straight off the shelf.


The idea is simple yet solid, even drawing intuitions from early research in image processing / Eigenfaces, which I can imagine inspiring future/new directions in representation learning.

**Weaknesses:**

I wonder about the scalability of PMAE when applied to larger images, with the potentially prohibitive costs of the eigendecomposition, e.g. $o(pixels^3)$. Could the method still perform well with low-rank approximations?

Some of the presented results are a bit unclear to me, see 'questions'.

**Questions:**

In the experiments exploring effects of masking ratios, as I understand it in the $PMAE_{rd}$ case, a random percentage of variance between $[10,90]$ is masked. I'm a bit confused as to why in Figure 5, which presents the impact of masking percentage, the classification accuracy results only shown for percentages between $[10, 50]$. What happens when a higher percentage of variance is masked?

How do you anticipate PMAE would perform with nonlinear transformations, e.g. kernelized PCA with an RBF kernel?

---

> ### Author Response · Authors · 2024-11-22
> **Rebuttal part 1**
>
> Dear reviewer WJuV,
>
> Thank you for your thorough review of our work!
> Please find below the answers to your questions.
>
> #### **Scalability of PMAE**
>
> As you correctly point out, the cost of eigendecomposition increases cubically with the data dimensionality, which poses challenges for very large datasets. Although computing a subset of principal components would help reduce these costs, its effects on the proposed masking strategy remain unclear. Indeed, performing a low-rank approximation would lead to a drop of the eigenvectors with lowest eigenvalues which previous work [1] showed carries features meaningful for perceptual tasks.
>
> Instead, future work could consider other data transformations like the Fourier transform [2], Wavelet transform [3], or Diffusion Maps [4] in place of the PCA. Notably, the Discrete Cosine Transform (DCT [5]), closely related to the Fourier transform, could be a viable cost-effective alternative to PCA.
> The DCT can be described as a discrete Fourier transform operating on real numbers only and is an approximation of PCA [6]. Its advantage lies in its cost. Unlike PCA, it relies on a fixed predefined basis and would thereby lower computational costs and help with scalability.
>
> Combined with our empirical findings on the benefits of the principal component space for image masking, these observations suggest that exploring a broader range of image transformations could open promising research directions while lowering computational costs.
>
> Thanks to your feedback and that of other reviewers, we have updated our section 8 to further discuss the scalability of PMAE to larger datasets as follows: _“Other off-the-shelf non-linear transformations, such as the Fourier transform, Wavelet transform, Kernel Principal Component Analysis, or Diffusion Maps, represent alternative candidate transformations. Future research should explore whether the properties of these spaces provide comparable or additional advantages over PCA. [...] A particularly appealing aspect of some of these methods (e.g., Fourier \& Wavelet transforms and Diffusion Maps) is the use of fixed bases, which could eliminate the computational overhead of PCA---whose cost scales cubically with the data dimensionality--- and improve scalability to larger datasets.”_
>
> [1] Balestriero, Randall, and Yann LeCun. "How Learning by Reconstruction Produces Uninformative Features For Perception." Forty-first International Conference on Machine Learning.
>
> [2] Bracewell, R. (1986). The Fourier Transform and Its Applications. McGraw-Hill.
>
> [3] Daubechies, I. (1992). Ten Lectures on Wavelets. SIAM.
>
> [4] Coifman, R. R., & Lafon, S. (2006). Diffusion Maps. Applied and Computational Harmonic Analysis, 21(1), 5–30.
>
> [5] Ahmed, Nasir, T_ Natarajan, and Kamisetty R. Rao. "Discrete cosine transform." IEEE transactions on Computers 100.1 (1974): 90-93.
>
> [6] Sanchez, Victoria, et al. "Diagonalizing properties of the discrete cosine transforms." IEEE transactions on Signal Processing 43.11 (1995): 2631-2641.

---

> > ### Author Response · Authors · 2024-11-22
> > **Rebuttal Part 2**
> >
> > #### **Larger range of parameters in masking ratio ablation**
> >
> > The complete range of hyperparameters was not included in the main manuscript due to space constraints. We have now included the full range ([10-90]) in Appendix A.6.1 of the updated manuscript. As shown in Figure 9, the performance continues to decrease for masking ratios larger than 50%.
> >
> > #### **PMAE with Kernel PCA**
> >
> > Thank you for the great suggestion! We are happy to see that you share our view that PMAE motivates the exploration of other off-the-shelf transformations for Masking Image Modelling. We found this suggestion interesting and relevant and performed preliminary experiments with kernel PCA with an RBF kernel (KMAE) on CIFAR10, for which results can be found below. The training pipeline was kept identical between PMAE and KMAE; the only modification is the replacement of PCA by kernel PCA for KMAE.
> >
> > Please note that scores for PMAE reported in Table 1 below are computed following the alternative objective described in “Additional results supporting our claims” in the general answer to reviewers.
> >
> >
> > |           | MAE   | PMAE* | KMAE* |
> > |-----------|-------|-------|-------|
> > | **Linear** | 50.7 | *59.0* | **64.0**  |
> > | **MLP**    | 55.2  | *64.1* | **68.6** |
> >
> > **Table 1**: Linear and MLP probe accuracy for MAE, PMAE, KMAE with best masking ratios for each method. Bold numbers refer to the best scores, italic to the second best. * refers to our work.
> >
> >
> > Table 1 reports the image classification accuracy using a linear and MLP probe for CIFAR10 for a standard MAE, PMAE, and KMAE, which relies on Kernel PCA. Results show performance gains of 13.3 and 5 percentage points brought by KMAE over MAE and PMAE, respectively, when using a linear probe. This proof of concept further highlights how spectral decomposition can offer an appealing partitioning of the information for Masked Image Modelling and shows how non-linear transformations could further enhance performance gains for downstream image classification. We have incorporated these results in Appendix A.6.4 and in our discussion in section 8.
> >
> > Thank you again for the great suggestion, which helped improve our work.
> >
> > We hope this answers your questions.
> > We greatly appreciate the time and effort you put into the review of our work and welcome any additional suggestions, questions, or requests for information.

---

### Author Response · Authors · 2024-11-22
**Rebuttal: General Answer to Reviewers**

Dear reviewers,

We thank you for the thorough reviews of our work and the comprehensive and valuable feedback. **We particularly value the acknowledgment that our work is impactful (_reviewer WJuv_), well-presented (_reviewers WJuv, tnEx,5bnU,1XJX_), well-motivated (_reviewer WJuv,1XJX_), that our idea is novel, intuitive, solid and interesting (_reviewers WJuv, tnEx,5bnU_), and that the proposed approach is supported by a solid experimental validation (_reviewer 5bnU_).**

Your detailed reviews have helped us identify areas for improvement and provided valuable insights that enhance the impact of our research. We have thoughtfully discussed your comments regarding the scalability of our approach (PMAE), and potential extensions to PMAE and have clarified some misunderstandings regarding PCA. We also used your feedback to extend the discussion section in our manuscript regarding our approach’s scalability. We have also included additional interesting results using kernel PCA as suggested by reviewer WJuv and provided a larger set of ablations over the masking ratio. **Please find these changes in blue font in our updated manuscript in section 8 and the appendix.**

---

> ### Author Response · Authors · 2024-11-22
> **Rebuttal: General Answer to Reviewers Part 2**
>
> #### **Scalability of PMAE** (_reviewers WJuv, tnEx,1XJX,5bnU_)
>
> Our work aims at showing the potential of the PC space for Masked Image Modeling (MIM) learning paradigms. We support our claims by showing the superiority of the space of principal components over the space of observations. We report substantial performance gains on 5 datasets with image resolutions ranging from 32x32 to 64x64 and training sample size ranging from 7k to 100k samples. These findings act as a proof of concept showing the appeal of PCA for MIM and suggest that exploring other image transformations (i.e., Fourier Transform [1], Wavelet transform [2], Diffusion Maps [3], Kernel PCA [6],...) is an interesting direction for further research.
>
> As PMAE relies on PCA, extending our experimental setup to larger-scale datasets requires the computation of their principal components. As rightfully mentioned by reviewer WJuv, the cost of this operation scales cubically with the data dimensionality (i.e., $O(d^2n+d^3)$ for a dataset with $n$ samples of dimensionality $d$).
>
> To lower this cost, future work could consider other data transformations like the Fourier transform [1], Wavelet transform [2], or Diffusion Maps [3]. Notably, the Discrete Cosine Transform (DCT [4]), closely related to the Fourier transform, could serve as a viable and cost-effective alternative to PCA. The DCT can be described as a discrete Fourier transform operating on real numbers only. In addition, the DCT is an approximation of PCA [5]. The advantage of the DCT lies in its cost. Unlike PCA, it relies on a fixed predefined basis and would lower computational costs and help scale up the proposed approach to even larger datasets.
>
> While PMAE is currently limited to mid-scale datasets, our work lays the groundwork for exploring new masking spaces for self-supervised learning by highlighting substantial performance gains obtained with PCA on various datasets.
>
> We have updated section 8 of the manuscript to further discuss the scalability of our approach as follows: _“Other off-the-shelf non-linear transformations, such as the Fourier transform, Wavelet transform, Kernel Principal Component Analysis, or Diffusion Maps, represent alternative candidate transformations. Future research should explore whether the properties of these spaces provide comparable or additional advantages over PCA. [...] A particularly appealing aspect of some of these methods (e.g., Fourier \& Wavelet transforms and Diffusion Maps) is the use of fixed bases, which could eliminate the computational overhead of PCA---whose cost scales cubically with the data dimensionality--- and improve scalability to larger datasets.”_
>
> We thank the reviewers for their valuable input, which helped improve our work.
>
> [1] Bracewell, R. (1986). The Fourier Transform and Its Applications. McGraw-Hill.
>
> [2] Daubechies, I. (1992). Ten Lectures on Wavelets. SIAM.
>
> [3] Coifman, R. R., & Lafon, S. (2006). Diffusion Maps. Applied and Computational Harmonic Analysis, 21(1), 5–30.
>
> [4] Ahmed, Nasir, T_ Natarajan, and Kamisetty R. Rao. "Discrete cosine transform." IEEE transactions on Computers 100.1 (1974): 90-93.
>
> [5] Sanchez, Victoria, et al. "Diagonalizing properties of the discrete cosine transforms." IEEE transactions on Signal Processing 43.11 (1995): 2631-2641.
>
> [6] Schölkopf, Bernhard, Alexander Smola, and Klaus-Robert Müller. "Kernel principal component analysis." International conference on artificial neural networks. Berlin, Heidelberg: Springer Berlin Heidelberg, 1997.

---

> > ### Author Response · Authors · 2024-11-22
> > **Rebuttal: General Answer to Reviewers Part 3**
> >
> > #### **Going beyond PCA** (_reviewer WJuv,5bnU_)
> >
> > Several reviewers have highlighted the potential of alternative transformations in addition to Principal Component Analysis (e.g., Fourier transform [1], Wavelet transform [2], Kernel PCA [3]). Reviewer WJuV noted that _“The idea is solid yet simple ... which I can imagine inspiring future/new directions in representation learning.”_.  We are pleased that the reviewers share our view that PMAE motivates the exploration of other off-the-shelf transformations for Masking Image Modelling. Thanks to reviewer WJuv’s suggestion, we have conducted preliminary experiments using Kernel PCA instead of PCA on the CIFAR10 dataset.
> >
> > Kernel PCA [3] is a non-linear data transformation, which differs from PCA by performing the spectral decomposition in a latent space in place of the space of observations. The data is first mapped to a high-dimensional space using a kernel function — we chose the Radial Basis Function (RBF) kernel — before spectral decomposition is performed. Table 1 below reports the image classification accuracy using a linear and MLP probe for CIFAR10 for a standard MAE, PMAE, and KMAE, which relies on Kernel PCA. Results show performance gains of 13.3 and 5 percentage points brought by KMAE over MAE and PMAE, respectively, when using a linear probe.
> >
> > These preliminary results highlight how spectral decomposition can offer an appealing partitioning of the information for MIM and show how non-linear transformations could further enhance performance gains for downstream image classification. These results were added to the manuscript and discussed in Appendix A.6.4. We thank reviewer WJuV for the great suggestion.
> >
> > Please note that scores for PMAE reported in table 1 below are computed following the alternative objective described in “Additional results supporting our claims” in the general answer to reviewers.
> >
> > |           | MAE   | PMAE* | KMAE* |
> > |-----------|-------|-------|-------|
> > | **Linear** | 50.7 | *59.0* | **64.0**  |
> > | **MLP**    | 55.2  | *64.1* | **68.6** |
> >
> > **Table 1**: Linear and MLP probe accuracy for MAE, PMAE, KMAE with best masking ratios for each method. Bold numbers refer to the best scores, italic to the second best. * refers to our work.
> >
> > [1] Bracewell, R. (1986). The Fourier Transform and Its Applications. McGraw-Hill.
> >
> > [2] Daubechies, I. (1992). Ten Lectures on Wavelets. SIAM.
> >
> > [3] Schölkopf, Bernhard, Alexander Smola, and Klaus-Robert Müller. "Kernel principal component analysis." International conference on artificial neural networks. Berlin, Heidelberg: Springer Berlin Heidelberg, 1997.

---

> > > ### Author Response · Authors · 2024-11-22
> > > **Rebuttal: General Answer to Reviewers Part 4**
> > >
> > > #### **Principal Components and Independence**
> > >
> > > We would like to address a misunderstanding raised by reviewer tnEx regarding a key aspect of our approach to ensure clarity and prevent further confusion.
> > >
> > > _“It does not make sense to me that we can recover removed principal components from other components. This is because unlike patches, these principal components are independent to each other.”_ - reviewer tnEx.
> > >
> > > Principal component analysis (PCA) finds components that maximize variance in the projected space, resulting in uncorrelated components (i.e., no linear relationship between principal components). However, this does not imply that the principal components are independent, as uncorrelatedness does not imply independence. In contrast, independent component analysis (ICA), which our work does not rely on, seeks to separate data into statistically independent components. In our opinion, it is reasonable to consider that natural or medical images are produced by a non-linear generative process. Consequently, it is unlikely that all principal components of natural or medical images would be independent.
> > >
> > > Our empirical results (c.f., table 1 in our manuscript)  show performance scores way above those of random predictions, which would not be possible if the input image and reconstruction target were to be independent. Finally, examples of visible and masked principled components, provided in Figure 3 (blue) of the original manuscript, also show visual evidence that disjoint sets of principal components are not independent of one another (i.e., the animal contours are present in both the input and target images in Figure 3 (blue)).
> > >
> > > We hope this dissipates any confusion regarding this fundamental aspect of our approach and are happy to discuss any follow-up questions.

---

> ### Author Response · Authors · 2024-11-22
> **Rebuttal: General Answer to Reviewers Part 5**
>
> #### **Additional results supporting our claims**
>
> Finally, we are pleased to report additional empirical results supporting our claims that the space of principal components is a meaningful masking space. We explore a simple change of our original setup, presented in Figure 1 of our manuscript, and observe larger performance gains compared to the results originally reported.
>
> In the original manuscript, we applied the reconstruction loss directly in the image space, as depicted in Figure 1. Our training objective, presented in equation 3.1, minimizes the Euclidean distance between the decoder’s output and the masked principal components projected back to image space. A simple alternative to this learning objective is to minimize the Euclidean distance in PC space between the masked principal components and the reconstruction of the masked principal components. This alternative to equation 3.1, now presented in appendix A.6.3 and equation A.1, further improves downstream classification performance as presented in tables 2 & 3 below.
>
> | Dataset       | MAE   | PMAE   |
> |---------------|-------|--------|
> | CIFAR10       | 50.7  | **59.0** |
> | TinyImageNet  | 15.5  | **22.5** |
> | BloodMNIST    | 78.6  | **95.5** |
> | DermaMNIST    | 73.7  | **78.6** |
> | PathMNIST     | 86.4  | **96.8** |
>
> **Table 2**: Linear probe accuracy for MAE, PMAE, for CIFAR10, TinyImageNet, BloodMNIST, DermaMNIST, PathMNIST for optimal masking ratios. Bold numbers refer to the best scores.
>
> | Dataset       | MAE   | PMAE   |
> |---------------|-------|--------|
> | CIFAR10       | 55.2  | **64.1** |
> | TinyImageNet  | 22.2  | **25.1** |
> | BloodMNIST    | 75.8  | **92.5** |
> | DermaMNIST    | 74.4  | **80.2** |
> | PathMNIST     | 95.1  | **98.6** |
>
> **Table 3**: MLP probe accuracy for MAE, PMAE, for CIFAR10, TinyImageNet, BloodMNIST, DermaMNIST, PathMNIST for optimal masking ratios. Bold numbers refer to the best scores.
>
> Tables 2 and 3 show an average performance gain of 9.6 percentage points over the MAE baseline with a linear probe. In comparison, our original results showed an average gain of 6.6 percentage points over the MAE baseline. These results further support our claims by showing that the modeling of masked principal components constitutes a strong learning paradigm.
>
> We welcome any additional suggestions, questions, or requests for information and encourage further discussions.
> Once again, we thank all the reviewers for their time and effort in evaluating our work.

---

### Author Response · Authors · 2024-11-25

Thank you once again for your constructive review of our work.

We have revised our submission to include additional results, integrated your feedback, and provided detailed responses to your questions below.

We are happy to engage in further discussion or address any additional questions you may have.
 If your questions have been addressed, we kindly hope you will consider raising your score.

---

### Comment · Reviewer_WJuv · 2024-11-27

I thank the authors for thoroughly addressing my questions and comments, as well as those of the other reviewers. I'm impressed by the improvement we see with kPCA, which further supports my positive evaluation that this angle, i.e. masking components in a latent space, is solid and can benefit the wider ICLR community.  I acknowledge the concerns raised by other reviewers, but I remain positive about this work and its potential and look forward to seeing where it goes next.

---

> ### Author Response · Authors · 2024-12-02
>
> We greatly appreciate your words of encouragement and your positive opinion of our work. Thank you again for your time and effort in reviewing our work!

---

### Meta-Review · Area_Chair_6h66 · 2024-12-20

**Metareview:**

This paper proposes a new idea for self-supervised visual learning. The conventional approach relies on training a model to predict the pixel values of randomly masked patches in training images. This paper proposes a radically different idea: instead of masking random patches, it masks one of the principal components (PCs) of the image and has the model predict the masked component from the remaining ones. The proposed method is evaluated by training a vision transformer architecture on TinyImageNet, CIFAR-10, and Medical MNIST, showing gains compared to patch-based masking.

While the idea is radically quite novel and interesting, and led to an engaging discussion between the authors and reviewers, these discussions reveal that the current draft needs improvement to become solid and convincing:

Reviewer tnEx finds the idea of predicting one PC from the rest somewhat problematic, as these bases capture orthogonal aspects of the image, making it difficult to predict data projected onto one PC from its projection onto the others. While the authors clarify that the PC bases are indeed orthogonal and that projecting data onto them decorrelates the data (making them linearly independent), they argue that the projected data can still contain nonlinear dependencies, which a nonlinear model like a neural network could potentially pick up. While true, it is still concerning that the approach is completely blind to first-order information in the data (linear relationships), which often contains significant information about the image and is necessary for its proper reconstruction.

Reviewer 5bnU questions what makes PCs a distinct and favored choice if one is already considering projecting data onto orthogonal bases.  There are infinitely many choices for such bases, with some having specific names and properties, such as Fourier bases. The reviewer wonders why PCs are the sensible choice for self-supervised visual learning. The authors respond that their framework can easily be extended to handle PCs in a latent space via the kernel trick, broadening the set of transformations for masked component learning. While true, this raises further questions about how to choose the best transform for learning masked components in self-supervised vision tasks. While a complete answer may be impossible, the question warrants further exploration of the choice of bases (including alternatives like Fourier), compared to what the paper currently offers.

Reviewer 1XJX questions the evaluation setting and why simple datasets are used to train a ViT architecture, resulting in significantly poor test accuracies. The authors point out they had difficulty applying their approach to larger datasets like ImageNet. This raised further concerns about the scalability of the approach. Computing PCs requires eigendecomposition or SVD, which is computationally expensive. While the authors emphasize that their work is merely a proof of concept and their goal is not large-scale experiments, there is no clear path to resolving the scalability issue. This raises the question of whether using PCs (which require SVD) is necessary, as opposed to alternative orthogonal bases like Fourier, which are much cheaper to compute.

In sum, while the paper considers a very interesting and novel approach to masking for self-supervised learning, there are several aspects that it can improve on to be considered a strong submission. I encourage the authors to continue pursuing this approach and resubmit their work after addressing the raised issues.

**Additional Comments On Reviewer Discussion:**

Reviewer tnEx finds the idea of predicting one PC from the rest somewhat problematic, as these bases capture orthogonal aspects of the image, making it difficult to predict data projected onto one PC from its projection onto the others. While the authors clarify that the PC bases are indeed orthogonal and that projecting data onto them decorrelates the data (making them linearly independent), they argue that the projected data can still contain nonlinear dependencies, which a nonlinear model like a neural network could potentially pick up. While true, it is still concerning that the approach is completely blind to first-order information in the data (linear relationships), which often contains significant information about the image and is necessary for its proper reconstruction.

Reviewer 5bnU questions what makes PCs a distinct and favored choice if one is already considering projecting data onto orthogonal bases.  There are infinitely many choices for such bases, with some having specific names and properties, such as Fourier bases. The reviewer wonders why PCs are the sensible choice for self-supervised visual learning. The authors respond that their framework can easily be extended to handle PCs in a latent space via the kernel trick, broadening the set of transformations for masked component learning. While true, this raises further questions about how to choose the best transform for learning masked components in self-supervised vision tasks. While a complete answer may be impossible, the question warrants further exploration of the choice of bases (including alternatives like Fourier), compared to what the paper currently offers.

Reviewer 1XJX questions the evaluation setting and why simple datasets are used to train a ViT architecture, resulting in significantly poor test accuracies. The authors point out they had difficulty applying their approach to larger datasets like ImageNet. This raised further concerns about the scalability of the approach. Computing PCs requires eigendecomposition or SVD, which is computationally expensive. While the authors emphasize that their work is merely a proof of concept and their goal is not large-scale experiments, there is no clear path to resolving the scalability issue. This raises the question of whether using PCs (which require SVD) is necessary, as opposed to alternative orthogonal bases like Fourier, which are much cheaper to compute.

---

### Decision · Program_Chairs · 2025-01-22

Reject